# Isotope engineering for spin defects in van der Waals materials

Ruotian Gong [1,7], Xinyi Du [1,7], Eli Janzen[2], Vincent Liu[3], Zhongyuan Liu[1], Guanghui He[1], Bingtian Ye[3], Tongcang Li [4,5], Norman Y. Yao[3], James H. Edgar [2], Erik A. Henriksen [1,6] & Chong Zu [1,6] ✉

Spin defects in van der Waals materials offer a promising platform for advancing quantum technologies. Here, we propose and demonstrate a powerful technique based on isotope engineering of host materials to significantly enhance the coherence properties of embedded spin defects. Focusing on the recently-discovered negatively charged boron vacancy center ($V_B^-$) in hexagonal boron nitride (hBN), we grow isotopically purified $h^{10}B^{15}N$ crystals. Compared to $V_B^-$ in hBN with the natural distribution of isotopes, we observe substantially narrower and less crowded $V_B^-$ spin transitions as well as extended coherence time $T_2$ and relaxation time $T_1$. For quantum sensing, $V_B^-$ centers in our $h^{10}B^{15}N$ samples exhibit a factor of 4 (2) enhancement in DC (AC) magnetic field sensitivity. For additional quantum resources, the individual addressability of the $V_B^-$ hyperfine levels enables the dynamical polarization and coherent control of the three nearest-neighbor $^{15}N$ nuclear spins. Our results demonstrate the power of isotope engineering for enhancing the properties of quantum spin defects in hBN, and can be readily extended to improving spin qubits in a broad family of van der Waals materials.

Optically-addressable spin defects in solid-state materials have emerged as one of the leading prospects for expanding the boundary of modern quantum technologies[1–17]. Recently, spin defects in atomically-thin van der Waals materials have attracted significant research interest for their innate ability to integrate with heterogeneous optoelectronic and nanophotonic devices[18–24]. Among a wide range of two-dimensional host materials, hexagonal boron nitride (hBN) stands out as a promising candidate owing to its large bandgap (~6 eV) and exceptional mechanical, thermal, and chemical stability[25,26]. Unlike established host materials such as diamond and silicon carbide, neither nitrogen nor boron has stable isotopes with zero nuclear spin, potentially to the detriment of spin defects in hBN. In particular, the presence of such a dense nuclear spin bath can substantially broaden electronic transitions and shorten the coherence time of spin defects.

To this end, prior experimental works have focused on designing dynamical decoupling sequences to isolate spin defects from the local nuclear spin environment in hBN[24,27].

In this work, we demonstrate isotope engineering of hBN as a more fundamental way of strengthening the capability of spin defects for quantum applications. By carefully optimizing the isotope species and preparing hBN samples with $^{15}N$ and $^{10}B$, we drastically improve the spin coherent properties of the recently discovered negatively charged boron-vacancy defect $(V_B^-)$[18,19]. We note that a prior study has investigated the effect of changing only boron isotopes in hBN and do not observe significant improvements in the spin properties of $V_B^-$ [28]. Importantly, this technique of isotope engineering is fundamental because of its full compatibility with other ongoing $V_B^-$ optimization efforts, making it a necessity for future $V_B^-$ improvement and

[1]Department of Physics, Washington University, St. Louis, MO 63130, USA. [2]Tim Taylor Department of Chemical Engineering, Kansas State University, Manhattan, KS 66506, USA. [3]Department of Physics, Harvard University, Cambridge, MA 02138, USA. [4]Department of Physics and Astronomy, Purdue University, West Lafayette, IN 47907, USA. [5]Elmore Family School of Electrical and Computer Engineering, Purdue University, West Lafayette, IN 47907, USA. [6]Institute of Materials Science and Engineering, Washington University, St. Louis, MO 63130, USA. [7]These authors contributed equally: Ruotian Gong, Xinyi Du. ✉e-mail: zu@wustl.edu

applications. Although here we focus on a specific spin defect in hBN, we would also like to highlight that the approach is exemplary for engineering other spin defects in general host materials.

## Results
### Optimal isotopes of hBN
We start with identifying the optimal isotope species in hBN for the embedded $V_B^-$ defects. The $V_B^-$ center harbors an electronic spin triplet ground state, $|m_s = 0, \pm 1\rangle$, which can be optically initialized and read out at room temperature[18]. The Hamiltonian of the system, including both the electronic degree of freedom and nearby nuclear spins, can be written as

$$H_{gs} = D_{gs}S_z^2 + \gamma_e B_z S_z - \sum_j \gamma_n^j B_z I_z^j + \sum_j \mathbf{S}\mathbf{A}^j\mathbf{I}^j, \quad (1)$$

where $D_{gs} = (2\pi) \times 3.48$ GHz is the ground state zero-field splitting between $|m_s = 0\rangle$ and $|m_s = \pm 1\rangle$, $B_z$ is the external magnetic field aligned along the out-of-plane c axis of hBN ($\hat{z}$ direction), $\mathbf{S}$ and $S_z$ are the electronic spin-1 operators, $\mathbf{I}^j$ and $I_z^j$ are the spin operators for the $j$th nuclear spin (including both nitrogen and boron) with hyperfine tensors $\mathbf{A}^j$, and $\gamma_e = (2\pi) \times 2.8$ MHz/G and $\gamma_n^j$ are the electronic and nuclear spin gyromagnetic ratios (Fig. 1a, c). We note that, for nuclei with $\mathcal{I} \geq 1$, there is a small additional nuclear spin dependent energy shift corresponding to higher multipole moments. Nevertheless, this term only acts on the nuclear spin degree of freedom and thus has no effect on the measured electronic spin transitions to leading order.

Throughout our experiment, we consistently operate within magnetic fields of $B_z \lesssim 760$ G such that the energy splitting $(D_{gs} \pm \gamma_e B_z) \gtrsim (2\pi) \times 1.35$ GHz between the electronic ground state levels $|m_s = 0\rangle$ and $|m_s = \pm 1\rangle$ is much greater than the hyperfine interaction strength, i.e. $|\mathbf{A}^j| \lesssim (2\pi) \times 100$ MHz. As a result, the hyperfine term can be approximated as $\sum_j \mathbf{S}\mathbf{A}^j\mathbf{I}^j \approx \sum_j A_{zz}^j S_z I_z^j = (\sum_j A_{zz}^j I_z^j)S_z$ to leading order, where all remaining terms are suppressed under the secular approximation. This term leads to an energy shift of the electronic spin transition that is dependent on the nuclear spin configuration; averaging across all different configurations gives rise to spectral crowding[29,30] of the electronic spin transition with an overall width proportional to $2|\gamma_n \mathcal{I}|$ (see Methods).

Figure 1a summarizes the nuclear spin quantum numbers and gyromagnetic ratios of the four stable atomic isotopes in hBN. Intuitively, a lower nuclear spin quantum number leads to fewer hyperfine

states and contributes to less spectral crowding, with all other factors being equal. Accordingly, the nitrogen isotope $^{15}$N ($\mathcal{I} = 1/2$) induces a narrower transition linewidth of $V_B^-$ than its naturally abundant counterpart, $^{14}$N ($\mathcal{I} = 1$). However, we note that the gyromagnetic ratio is equally important in isotope engineering, and although $^{11}$B ($\mathcal{I} = 3/2$) has a lower spin quantum number compared to $^{10}$B ($\mathcal{I} = 3$), its gyromagnetic ratio is much larger, which results in a broader linewidth. Therefore, to minimize the spectral crowding of $V_B^-$, we predict that h$^{10}$B$^{15}$N is the optimal host material.

### Experimental characterization
To experimentally validate the effect of isotope engineering on the $V_B^-$ center, we grow single h$^{10}$B$^{15}$N crystals with isotopically purified $^{15}$N and $^{10}$B sources (purity > 99.7% for $^{15}$N and > 99.2% for $^{10}$B, see Methods)[31–34]. We exfoliate the hBN crystal into thin flakes with thicknesses ranging from 20 to 70 nm. $V_B^-$ defects are created via He$^+$ ion implantation with energy 3 keV and dosage 1 ion/nm$^2$, resulting an estimated $V_B^-$ concentration around 150 ppm[24]. In the experiment, the fluorescence signal of $V_B^-$ is collected using a home-built confocal microscope, and the microwave is delivered via a coplanar waveguide (see Supplementary Note 1). Most of the experiment in this work are performed at room temperature other than that specified (Fig. 2c).

The spin transitions of $V_B^-$ can be probed via electron spin resonance (ESR) spectroscopy: by sweeping the frequency of the applied microwave drive while monitoring the fluorescence signal of $V_B^-$, we expect a fluorescence drop when the microwave is resonant with an electronic spin transition. Figure 1c compares the ESR spectra of $V_B^-$ in our isotopically engineered h$^{10}$B$^{15}$N and conventional naturally abundant hBN$_{nat}$ (99.6%/0.4% for $^{14}$N/$^{15}$N and 20%/80% for $^{10}$B/$^{11}$B) under a small external magnetic field ($B \approx 87$ G). As expected, the ESR measurements for both $|m_s = 0\rangle \leftrightarrow |m_s = \pm 1\rangle$ transitions on h$^{10}$B$^{15}$N show much less crowded spectra than hBN$_{nat}$. In particular, there are two striking features. First, instead of an ordinary 7-resonance spectrum observed in the naturally abundant sample[18,35], we resolve 4 distinct hyperfine lines for $V_B^-$ in h$^{10}$B$^{15}$N. This structure stems from the hyperfine interaction between $V_B^-$ and the three nearest-neighbor $^{15}$N nuclear spins ($\sum_{j=1}^3 A_{zz}^{15}{}^N I_z^j)S_z$, where $I_z^j$ can take the values of $\pm\frac{1}{2}$. Accounting for all nuclear spin configurations, there are a total of 4 hyperfine lines (split by $|A_{zz}^{15N}|$) with degeneracy {1:3:3:1} corresponding to the total nuclear spin number $\sum m_I = \{-\frac{3}{2}, -\frac{1}{2}, \frac{1}{2}, \frac{3}{2}\}$ (Fig. 1b, c). By fitting the spectrum to the sum of four equally-spaced Lorentzians with amplitudes proportional to the aforementioned degeneracy ratio, we

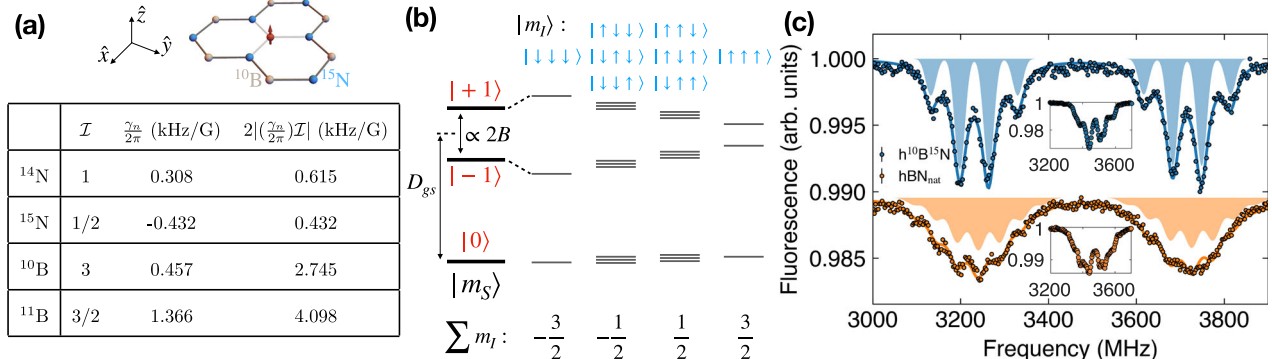

**Fig. 1 | $V_B^-$ electron spin resonance spectra in isotopically distinct hBN samples. a** Schematic of an individual $V_B^-$ center (red spin) in an isotopically purified h$^{10}$B$^{15}$N crystal. $\hat{z}$ is defined along the c-axis (perpendicular to the lattice plane) and $\hat{x}$ and $\hat{y}$ lie in the lattice plane, with $\hat{x}$ oriented along one of the three $V_B^-$ Nitrogen bonds. The table lists the nuclear spin quantum number ($\mathcal{I}$), gyromagnetic ratio ($\gamma_n$), and overall electronic spin transition width ($\propto 2|\gamma_n \mathcal{I}|$) of the four stable atomic isotopes in hBN. **b** Energy level diagram of the $V_B^-$ electronic ground state coupled to the three nearest-neighbor $^{15}$N nuclear spins ($\mathcal{I} = \frac{1}{2}$) under an external magnetic

field $B_z$. For each electronic spin transition $|m_s = 0\rangle \leftrightarrow |m_s = \pm 1\rangle$, the hyperfine interaction leads to four distinct resonances with total nuclear spin magnetic quantum number $\sum m_I = \{-\frac{3}{2}, -\frac{1}{2}, \frac{1}{2}, \frac{3}{2}\}$ of degeneracy {1, 3, 3, 1}. **c** Measured ESR spectra of $V_B^-$ in h$^{10}$B$^{15}$N and naturally abundant hBN$_{nat}$ at magnetic field $B_z \approx 87$ G. Solid lines represent multi-peak Lorentzian fits, and shaded regions represent numerically simulated transitions considering the 36 nearest nuclear spins (18 nitrogen and 18 boron atoms; see Methods). Insets: ESR spectra at $B_z = 0$ G. Source data are provided as a Source Data file.

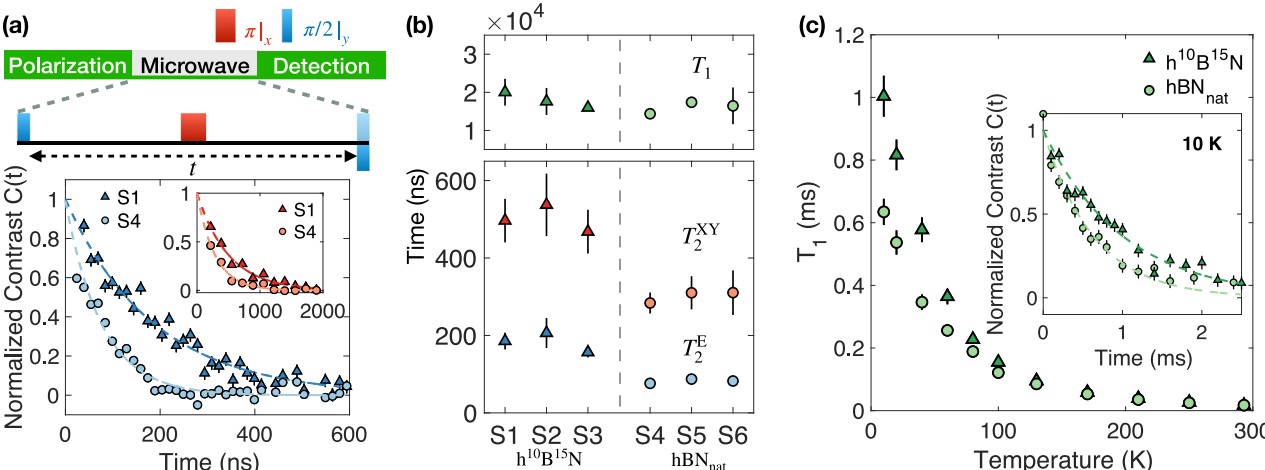

**Fig. 2 | Coherent dynamics of $V_B^-$ in isotopically distinct hBN samples. a** $T_2^E$ spin echo measurements on samples S1 ($h^{10}B^{15}N$) and S4 ($hBN_{nat}$). Insets: $T_2^{XY}$ XY8 measurements on samples S1 and S4. Error bars represent 1 s.d. accounting statistical uncertainties. **b** Bottom: extracted spin coherence timescales $T_2^E$ and $T_2^{XY}$. Top: extracted spin relaxation timescales $T_1$ for all six hBN samples investigated in this work. Error bars in time represent 1 s.d. accounting fitting error. **c** Relaxation timescales $T_1$ for both $h^{10}B^{15}N$ and $hBN_{nat}$ under different temperatures. Inset: $T_1$ measurement comparison with temperature at 10 K. Error bars represent 1 s.d. accounting statistical uncertainties, and error bars in $T_1$ represent 1 s.d. accounting fitting error. Source data are provided as a Source Data file.

extract the hyperfine coupling $A_{zz}^{15N} = (2\pi) \times [-65.9 \pm 0.9]$ MHz, where the negative sign originates from the negative gyromagnetic ratio of $^{15}$N. In comparison, the nearest-neighbor $^{14}$N hyperfine interaction strength in $hBN_{nat}$ is $A_{zz}^{14N} = (2\pi) \times [48.3 \pm 0.5]$ MHz. The measured ratio $A_{zz}^{15N}/A_{zz}^{14N} \approx -1.36$ agrees well with the nuclear spin gyromagnetic ratios $\gamma_n^{15N}/\gamma_n^{14N} \approx -1.4$.

The second feature of the $V_B^-$ spectrum in $h^{10}B^{15}N$ is its dramatically narrower transitions. Specifically, the full width at half maximum (FWHM) linewidth for each hyperfine resonance is $(2\pi) \times [55 \pm 2]$ MHz, almost a factor of two lower than that of the naturally abundant $hBN_{nat}$ sample, $(2\pi) \times [95 \pm 2]$ MHz. The linewidth originates from the hyperfine couplings to boron and nitrogen nuclear spins beyond the three nearest-neighbor atoms. To quantitatively capture the measured ESR linewidth, we perform numerical simulations by summing over the couplings to the nearest 36 nuclear spins for four different kinds of isotopically distinct hBN samples (see Methods and Supplementary Information)[36]. The simulated spectra for $h^{10}B^{15}N$ and $hBN_{nat}$ are in good agreement with the experimental data (Fig. 1c). Importantly, the narrower and less crowded hyperfine lines in $h^{10}B^{15}N$ substantially boost the sensitivity of $V_B^-$ for quantum sensing applications[37].

## Enhanced quantum sensing

Taking magnetic field sensing as an example, we now evaluate the DC and AC magnetic field sensitivity enhancement of $V_B^-$ in $h^{10}B^{15}N$. The static magnetic field sensitivity of ESR measurement takes the form[38,39]

$$\eta_{DC} \approx \frac{2\pi}{\gamma_e \sqrt{R}} \left( \max \left| \frac{\partial C(\nu)}{\partial \nu} \right| \right)^{-1} \approx \frac{8\pi}{3\sqrt{3}} \frac{1}{\gamma_e} \frac{\Delta\nu}{C_m \sqrt{R}}, \quad (2)$$

where $R$ denotes the photon detection rate, $C(\nu)$ the ESR measurement contrast at microwave frequency $\nu$, $C_m$ the maximum contrast, and $\Delta\nu$ the FWHM linewidth assuming a single Lorenztian resonance. If one directly compares the fitted FWHM from $h^{10}B^{15}N$ and $hBN_{nat}$, we find a factor of ~1.8 improvement in sensitivity. Moreover, the hyperfine transitions of $V_B^-$ in $hBN_{nat}$ significantly overlap with each other, while for $V_B^-$ in $h^{10}B^{15}N$ the resonances are individually resolvable. Therefore, a more accurate comparison instead takes the steepest slope, $\max \left| \frac{\partial C(\nu)}{\partial \nu} \right|$, into consideration, with which we conclude a factor of ~4 improvement in sensitivity using isotope purified $h^{10}B^{15}N$. By optimizing the laser and microwave power, we estimate a static magnetic field sensitivity $\eta_{DC} \approx 10$ μT Hz$^{-\frac{1}{2}}$. We note that the sensitivities reported in

this work are sample specific, and one can further optimize the sensitivity by increasing the number of $V_B^-$ defects used in experiment (see Supplementary Note 4).

Next, we characterize the spin echo coherence of $V_B^-$ in $h^{10}B^{15}N$ and $hBN_{nat}$ for AC field sensing. The spin echo coherent timescale, $T_2^E$, of $V_B^-$ has been previously understood to be limited by the magnetic fluctuations from the nearby nuclear spin bath[24,28,40], and the isotope engineering technique should precisely alleviate such decoherence. For a robust and systematic comparison, we prepare three $h^{10}B^{15}N$ flakes (sample S1-S3) and three naturally abundant $hBN_{nat}$ flakes (sample S4-S6) onto the same microwave stripline (see Methods). Figure 2a shows the measured spin echo decays on $h^{10}B^{15}N$ sample S1 and $hBN_{nat}$ sample S4, where the extension of spin echo timescales $T_2^E$ exceeds twofold, from $T_{2,S4}^E = 87 \pm 9$ ns to $T_{2,S1}^E = 186 \pm 22$ ns. Figure 2b summarizes the experimentally measured spin echo coherence timescales $T_2^E$ across all six hBN flakes, revealing a consistent improvement of coherence time in our isotopically purified samples. We also perform the spin echo measurement using $h^{11}B^{15}N$ samples and find $T_2^E \approx 119 \pm 6$ ns, lying in between $h^{10}B^{15}N$ and $hBN_{nat}$ (see Supplementary Fig. 13).

The AC magnetic field sensitivity $\eta_{AC} \propto T_2^{-1}$ exhibits improvement of more than a factor of 2 in our $h^{10}B^{15}N$ samples. By employing an advanced dynamical decoupling sequence, XY8, which exploits a series of echo pulses with alternating phases, we further extend the coherence time of $V_B^-$ in $h^{10}B^{15}N$ samples to $T_2^{XY} \approx 500$ ns (Fig. 2b Inset). This improves the estimated AC magnetic field sensitivity to $\eta_{AC} \approx 7$ μT Hz$^{-\frac{1}{2}}$ for a signal at frequency ~$(2\pi) \times 12$ MHz (see Supplementary Note 4).

We now turn to investigate the spin relaxation time, $T_1$, of $V_B^-$ in different isotopic samples. At room temperature, we find $T_1$ of $V_B^-$ across all six samples to be comparable ($T_1 \approx 15$ μs) and nearly independent of isotope choice (Fig. 2b). To further explore the isotope effect on $T_1$, we also explore the temperature-dependence of the $V_B^-$ relaxation using an optical cryostat (Fig. 2c). The measured $T_1$ in both $h^{10}B^{15}N$ and $hBN_{nat}$ increases monotonically with decreasing temperatures. Interestingly, when the temperature goes below ~170 K, there is a clear improvement of $T_1$ in the $h^{10}B^{15}N$ sample compared to naturally abundant sample. At the lowest temperature 10 K, we find $T_1 = (1.00 \pm 0.07)$ ms for $V_B^-$ in $h^{10}B^{15}N$, around 50% longer than $T_1 = (0.63 \pm 0.04)$ ms measured in $hBN_{nat}$ (Fig. 2c Inset). The current theoretical understanding of the spin relaxation process of $V_B^-$

identifies spin-phonon interaction as the main limitation[19,41,42], which is consistent with the observed monotonic increase of $T_1$ with decreasing temperature in both samples. One possible explanation for the improved $T_1$ of $V_B^-$ in h$^{10}$B$^{15}$N is that the nearest-neighboring $^{15}$N nuclei are slightly heavier in mass, leading to a weaker spin-phonon coupling strength[43,44]. However, the detailed underlying mechanism of isotope and temperature dependences of $V_B^-$ spin lifetime invites more future studies. Nevertheless, the extension of $T_1$ from isotope effect facilitates the use of $V_B^-$ as a noise magnetometer for diagnosing different phases of material, such as magnetic insulators and superconductivity at low temperature[14,45–47].

### Polarizing three nearest $^{15}$N Nuclear spins

Next, we demonstrate the polarization and coherent control of the three nearest-neighbor $^{15}$N nuclear spins using $V_B^-$. Nuclear spins feature exceptional isolation from external environments and offer long-lived systems for quantum simulation and computation applications. In contrast to $V_B^-$ in naturally abundant hBN samples, our isotopically purified host h$^{10}$B$^{15}$N presents a unique advantage: the hyperfine levels are individually addressable with a much less crowded spectrum.

We start by dynamically polarizing the $^{15}$N nuclear spins at the electronic spin level anti-crossing (esLAC). Under an external magnetic field $B_z \approx 760$ G, the $V_B^-$ excited state levels $|m_s = 0\rangle$ and $|m_s = -1\rangle$ are nearly degenerate[48–50]. In this regime, the secular approximation no longer holds, and we need to consider the full hyperfine interaction Hamiltonian,

$$\sum_{j=1}^{3} \mathbf{S} \mathbf{A}^j \mathbf{I}^j = \sum_{j=1}^{3} (A_{zz}^j S_z I_z^j + A_{xx}^j S_x I_x^j + A_{yy}^j S_y I_y^j + A_{xy}^j S_x I_y^j + A_{yx}^j S_y I_x^j). \quad (3)$$

To understand the nuclear spin polarization process, we can rewrite the hyperfine terms using ladder operators as

$$\sum_{j=1}^{3} \mathbf{S} \mathbf{A}^j \mathbf{I}^j = \sum_{j=1}^{3} [A_{zz}^j S_z I_z^j + (A_1^j S_+ I_-^j + h.c.) + (A_2^j S_+ I_+^j + h.c.)], \quad (4)$$

where $A_1^j = \frac{1}{4}(A_{xx}^j + A_{yy}^j)$, $A_2^j = \frac{1}{4}(A_{xx}^j - A_{yy}^j) + \frac{1}{2i} A_{xy}^j$, $S_\pm$ and $I_\pm$ are the ladder operators (see Supplementary Note 5).

From Eq. (4), the nuclear spin polarization process at esLAC is made apparent. Specifically, the term $(A_1^j S_+ I_-^j + h.c.)$ leads to electron-nuclear spin flip-flop, $|m_s = 0, m_I = \downarrow\rangle \leftrightarrow |m_s = -1, m_I = \uparrow\rangle$, while the term $(A_2^j S_+ I_+^j + h.c.)$ connects the other two states, $|m_s = 0, m_I = \uparrow\rangle \leftrightarrow |m_s = -1, m_I = \downarrow\rangle$ (Fig. 3a). From previous ab-initio calculations[35,36], $|A_1^j| > |A_2^j|$ for the three nearest-neighbor $^{15}$N nuclear spins. Therefore, under the strong optical polarization that continuously pumps the electronic spin from $|m_s = \pm 1\rangle$ to $|m_s = 0\rangle$ (while leaving the nuclear spin unchanged), each $^{15}$N will be preferentially polarized to $|m_I = \uparrow\rangle$.

To experimentally probe the nuclear spin polarization, we measure ESR spectra at a range of different laser powers. Here, we focus on the ESR transitions from $|m_s = 0\rangle$ to $|m_s = +1\rangle$ to avoid any fluorescence modulation due to esLAC and thus accurately characterize the nuclear spin population[51]. At low laser power (0.3 mW), the ESR spectrum exhibits a symmetric four-peak structure with amplitudes {1:3:3:1}, similar to the low field measurement, indicating no nuclear spin polarization (Fig. 3b). Once the laser power, and hence the optical pumping rate, increases, the ESR spectrum manifests an asymmetry skewed toward the two resonances at lower frequencies. This indicates that the three nearest-neighbor $^{15}$N nuclear spins are significantly polarized to the $|m_I = \uparrow\rangle$ state.

To quantify the nuclear spin polarization, we assume each nuclear spin is independently initialized to $|m_I = \uparrow\rangle$ with probability $P(|\uparrow\rangle)$ and $|m_I = \downarrow\rangle$ with probability $1 - P$. In this case, the population distribution of the four hyperfine resonances follows a ratio of $\{P^3 : 3P^2(1-P) : 3P(1-P)^2 : (1-P)^3\}$ (See Methods). This model yields excellent agreement with the experimental data (Fig. 3b), allowing us to extract the nuclear spin polarization probability $P$ at different laser powers. The nuclear polarization saturates at a laser power of around 10 mW, giving a maximum extracted nuclear spin polarization probability $P = (63.2 \pm 0.3)\%$ (Fig. 3c). The nuclear spin polarization is also sensitive to the magnetic field alignment relative to the c-axis of the h$^{10}$B$^{15}$N, as a transverse magnetic field induces mixing between the nuclear spin states (Fig. 3c inset). We notice a small overall shift (~5 MHz) of the entire ESR spectrum at high laser power originating from the laser-induced heating effect of the sample[37]. Nevertheless, the well-resolved hyperfine lines in our isotopically purified h$^{10}$B$^{15}$N enable us to faithfully capture any temperature-induced shift that may have been previously difficult to distinguish in the spectrum of naturally abundant hBN$_{nat}$.

We further study the nuclear spin depolarization dynamics via pulsed ESR measurements. After initializing the $V_B^-$ electronic spin and the three $^{15}$N nuclear spins at esLAC with a 5 μs laser pulse, we wait for a time $t$ before applying a microwave $\pi$-pulse with varying frequency to probe the hyperfine transitions, as shown in Fig. 3d. By fitting the ESR spectra, we observe that the nuclear spin population $P$ exhibits almost no decay within 15 μs, the $T_1$ time of $V_B^-$. The longevity of nuclear spin polarization may serve as a vital attribute for future applications such as quantum registers[52].

### Coherent control of the nuclear spins

Next, we illustrate the direct control and detection of the three nearest-neighbor $^{15}$N nuclear spins. After polarizing the $V_B^-$ to $|m_s = 0\rangle$ state via optical pumping, we apply a nuclear spin selective microwave $\pi$-pulse to transfer $|m_s = 0, \sum m_I = \frac{1}{2}\rangle$ population into $|m_s = -1, \sum m_I = \frac{1}{2}\rangle$ followed by a radio-frequency (RF) $\pi$-pulse to drive the nuclear spin states (Fig. 4a). A final microwave $\pi$-pulse transfers $|m_s = -1, \sum m_I = \frac{1}{2}\rangle$ back to $|m_s = 0, \sum m_I = \frac{1}{2}\rangle$ for nuclear spin detection.

Figure 4b shows the measured $^{15}$N nuclear spin resonance spectrum obtained by sweeping the frequency of the RF pulse. The clear nuclear spin transition at around $(2\pi) \times 66.6$ MHz is consistent with the sum of the measured $^{15}$N hyperfine interaction strength and the nuclear spin Zeeman shift $|A_{zz}^{15N} + \gamma_n B_z| \approx (2\pi) \times 66.2$ MHz. By parking the RF pulse at this frequency and increasing the pulse duration, we obtain coherent Rabi oscillations of the nearest three $^{15}$N nuclear spins with Rabi frequency $\Omega_N = (2\pi) \times (1.67 \pm 0.02)$ MHz (Fig. 4c).

Here, we would like to emphasize a few points. First, the nuclear spin Rabi signal contains a clear beating that does not fit to a single frequency oscillation. This originates from the interaction between nuclear spins which can be captured using a more sophisticated model containing $V_B^-$ and three $^{15}$N nuclear spins (see Supplementary Note 6.3). Second, the transverse hyperfine interaction enhances the effective nuclear gyromagnetic ratio to[35,53]

$$\gamma_n^{\text{eff}} \approx \frac{\gamma_e \sqrt{A_{xx}^2 + A_{yy}^2 + 2A_{xy}^2}}{\sqrt{2}(D_{gs} - \gamma_e B_z)}, \quad (5)$$

enabling us to achieve a fast nuclear spin manipulation (see Supplementary Note 6.1). From our experiment, we estimate $\gamma_n^{\text{eff}}/\gamma_n \approx 99$, suggesting $\sqrt{A_{xx}^2 + A_{yy}^2 + 2A_{xy}^2} \approx (2\pi) \times 30$ MHz, a factor approximately 5 times lower than the values predicted by ab-initio calculations[35,36] (see Supplementary Note 6.2). Another piece of evidence is that the measured nuclear spin resonance spectrum is consistent with the estimated transverse hyperfine parameters (Fig. 4b), while if we include the ab-initio predicted values, the simulated spectrum deviates from the measurement at $B \approx 760$ G (Supplementary Fig. 9). Prior work using $^{14}$N nuclear spins in naturally abundant hBN$_{nat}$ has reported a ~30% discrepancy between experiment and the ab-initio calculated transverse hyperfine coefficients[35]. However, the measured $^{14}$N nuclear spin transitions contain multiple resonances and are much broader

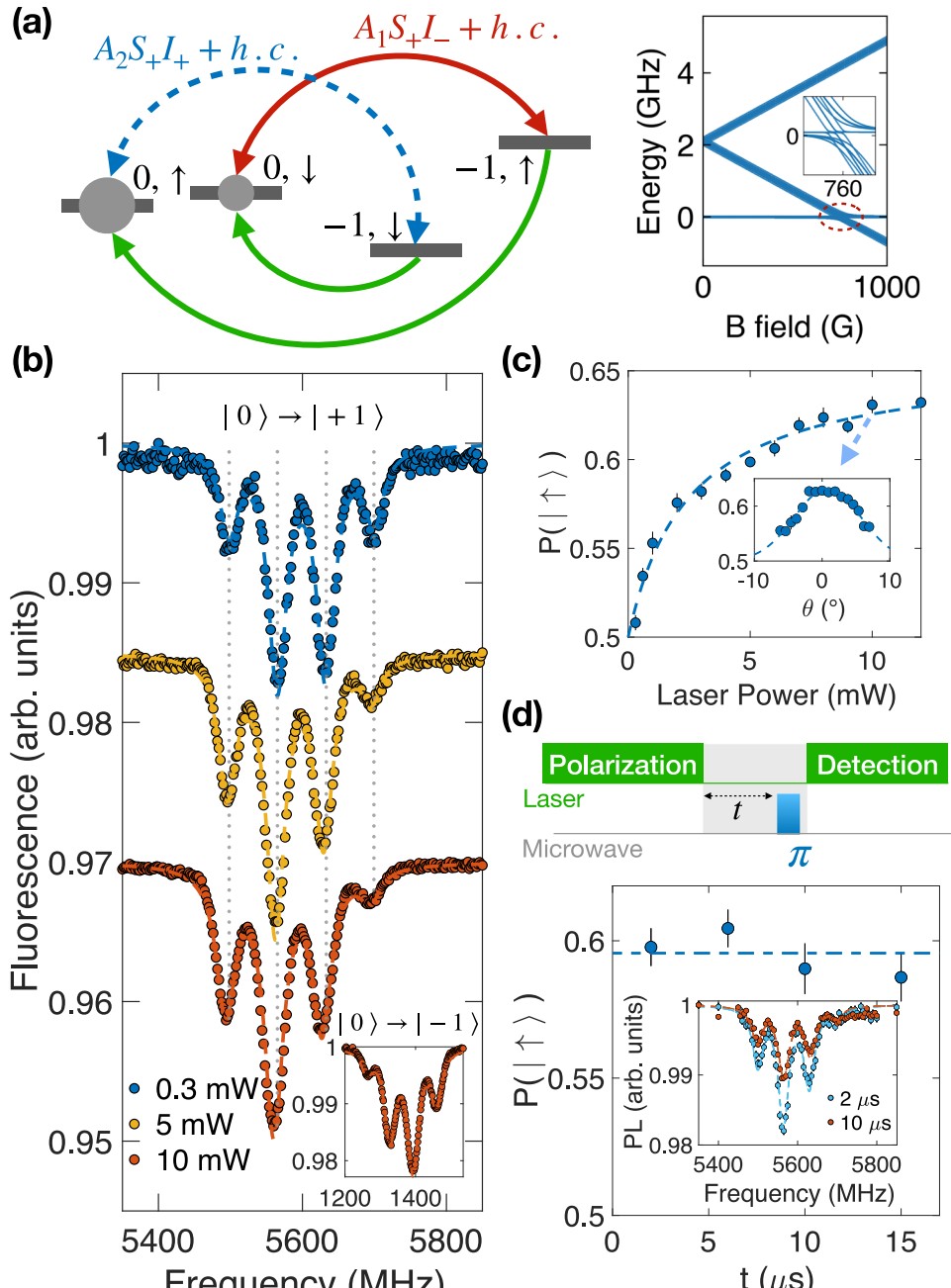

**Fig. 3 | Dynamic polarization of nearest-neighbor $^{15}$N nuclear spins. a** Left: schematic of the nuclear spin polarization process of $^{15}$N spins at esLAC ($B_z = 760$ G). The hyperfine term ($A_1 S_+ I_- + h.c.$) (red) is much stronger than ($A_2 S_+ I_+ + h.c.$) (blue). Under laser pumping (green), which continuously polarizes the electronic spin to $|m_s = 0\rangle$, each nuclear spin will be preferentially polarized to $|m_I = \uparrow\rangle$. Right: calculated excited state energy spectrum of $V_B^-$ as a function of magnetic field (see Supplementary Note 2). Inset: closeup of energy levels around esLAC (760 G). **b** Measured ESR spectra of the $|m_s = 0\rangle \leftrightarrow |m_s = +1\rangle$ transition at esLAC under different laser powers. Inset: ESR spectrum of the $|m_s = 0\rangle \leftrightarrow$ $|m_s = -1\rangle$ transition at a laser power of ~10 mW. Dashed lines represent best fits assuming each $^{15}$N nucleus is independently polarized to $|\uparrow\rangle$ with probability $P$. **c** Extracted $P(|\uparrow\rangle)$ as a function of laser power. Inset: $P(|\uparrow\rangle)$ as a function of the magnetic field alignment angle $\theta$ relative to the c-axis of hBN. Error bars represent 1 s.d. accounting fitting error. **d** Measured depolarization dynamics of nuclear spin as a function of wait time $t$. Inset: pulsed-ESR spectra at $t = 2\,\mu s$ and $10\,\mu s$, where PL is the fluorescence signal. Error bars represent 1 s.d. accounting fitting error. Source data are provided as a Source Data file.

than what we have observed here using $^{15}$N nuclei. Future work is required to resolve such discrepancy between experiment and ab-initio results. For instance, one can perform high-resolution ESR spectroscopy near the ground-state anti-crossing (gsLAC) with magnetic field $B \approx 1200$ G. At such field, the $V_B^-$ electronic spin state $|0\rangle$ and $|-1\rangle$ are close to degenerate, and one expects the effect of transverse hyperfine coefficients to become evident.

## Outlook

In conclusion, we present isotope engineering as a versatile and foundational tool to improve spin properties of $V_B^-$. On the quantum sensing front, $V_B^-$ centers in isotope-engineered h$^{10}$B$^{15}$N feature substantially narrower spin transitions and enhanced spin coherent time compared to conventional hBN$_{nat}$, enabling a universal enhancement factor of ~4 (2) in DC (AC) magnetic field sensitivity. Additionally, we

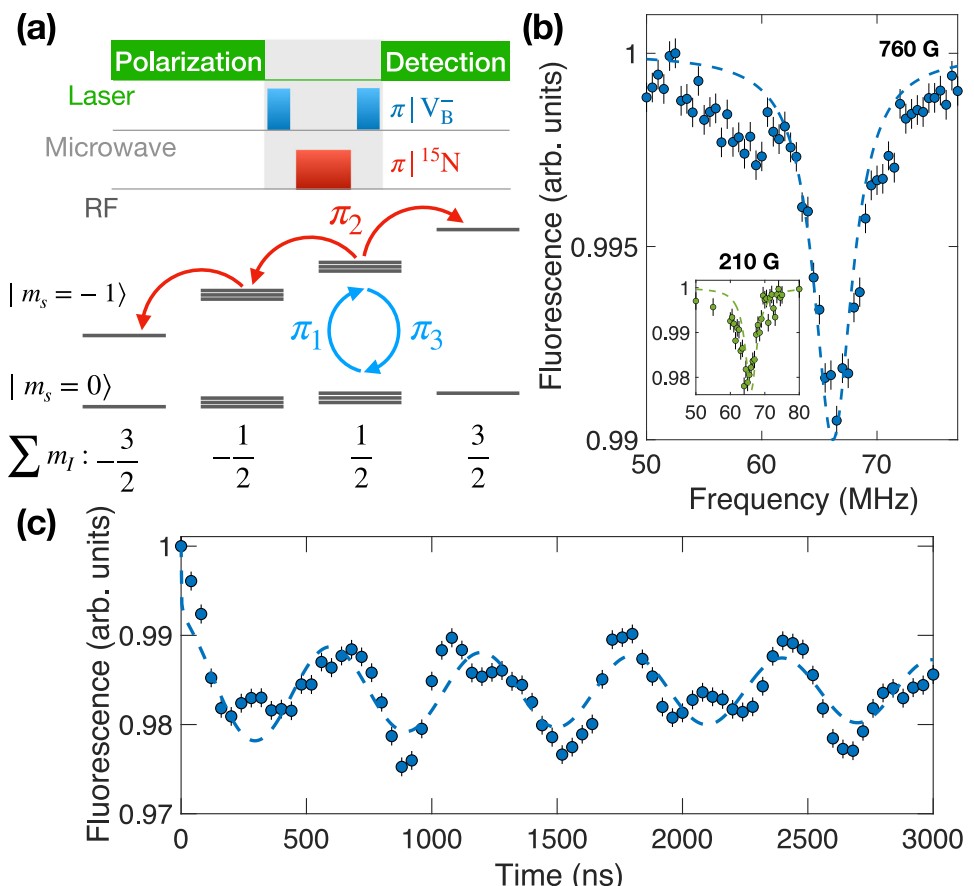

**Fig. 4 | Coherent control of $^{15}$N nuclear spins. a** Experimental pulse sequence and schematic for the control and detection of nuclear spins. **b** Resulting $^{15}$N nuclear spin resonance spectra at 760 G and 210 G (Inset). Dashed lines represent simulated spectra using adjusted transverse hyperfine terms (see Supplementary Note 6.3). Error bars represent 1 s.d. accounting statistical uncertainties. **c** Measurement of nuclear spin Rabi oscillations. The dashed line corresponds to a cosine fit with exponentially decaying amplitude that takes the form $Ae^{-(t/T)^{\alpha}} \cdot \cos(2\pi t/\Omega) + c$. Error bars represent 1 s.d. accounting statistical uncertainties. Source data are provided as a Source Data file.

demonstrate the dynamical polarization and coherent control of the nearest neighboring three $^{15}$N nuclear spins.

Looking forward, our work also opens the door to a few highly promising directions. First, we believe that *almost all* previous works using $V_B^-$ for sensing applications could greatly benefit from simply switching to our newly prepared h$^{10}$B$^{15}$N sample. The demonstrated sensitivity improvement is also fully compatible with other ongoing optimization efforts on $V_B^-$ sensors, such as dynamical decoupling protocols to extend $T_2$[24,27], coupling $V_B^-$ to optical cavities[54–56] and increasing $V_B^-$ number and density[24,57,58] to improve fluorescent signals. Second, the individual addressability of $V_B^-$ hyperfine transitions in h$^{10}$B$^{15}$N allows the utilization of nearby nuclear spins as valuable quantum resources and provides a promising platform to realize multiqubit quantum registers in two-dimensional materials. Finally, we highlight that our isotope engineering method can be readily extended to other spin defects in hBN[59,60] as well as the broader family of van der Waals materials[61]. For instance, the recent search for quantum spin defects in transition metal dichalcogenides[62] will greatly benefit from a careful selection of host isotopes, which may even create a nuclear spin-free environment.

## Methods

### hBN device fabrication

**Growth of isotopically purified h$^{10}$B$^{15}$N.** High-quality h$^{10}$B$^{15}$N flakes are grown via precipitation from a molten Ni-Cr flux as detailed in a separate work[34]. Ni, Cr, and isotopically pure $^{10}$B powders in the mass ratio 12:12:1 [Ni:Cr:B] are first loaded into an alumina crucible, then heated at 1550 °C for 24 hr in a continuously flowing Ar/H$_2$ atmosphere to melt all three components into a homogenous ingot and remove oxygen impurities. Next, this ingot is heated at 1550 °C for 24 hr in a static $^{15}$N$_2$/H$_2$ atmosphere to saturate the molten flux with $^{15}$N, then slowly cooled at 1°C/hr to precipitate hBN from the solution. Since isotopically pure $^{10}$B (>99.2%) and $^{15}$N (>99.7%) are the only sources of boron and nitrogen in the system, the hBN single crystals that precipitated have the same isotopes.

**Creation of $V_B^-$ defects.** We first exfoliate hBN nanosheets using tapes and then transfer them onto the Si/SiO$_2$ wafer. The wafer is pretreated with O$_2$ plasma at 50 W for 1 min (flow rate 20sccm). The tapes and wafers are heated to 100 °C for 1 min before samples are carefully peeled off for maximum hBN flake yield[63]. The wafers with hBN$_{nat}$ and h$^{10}$B$^{15}$N are then sent to CuttingEdge Ions LLC for $^4$He$^+$ ion implantation with energy 3 keV and dose density 1ion/nm$^2$ to create $V_B^-$ defects with estimated density ~150 ppm[24]. After implantation, the hBN flakes with thickness ranging from 20 to 70 nm are transferred onto the coplanar waveguide using polycarbonate (PC) stamps. Specifically, the temperature is raised in a stepwise manner to facilitate the transfer of hBN flakes. After successful transfer, the waveguide is immersed in chloroform for cleaning, removing any residual PC films. For a systematic comparison between $V_B^-$ in isotopically purified h$^{10}$B$^{15}$N and naturally abundant hBN$_{nat}$, we transfer three h$^{10}$B$^{15}$N flakes (S1-S3) and three hBN$_{nat}$ flakes (S4-S6) onto a single waveguide for measurement (Supplementary Fig. 1).

**Fabrication of Coplanar Waveguide.** An impedance-matched (50 Ω) coplanar waveguide is fabricated onto a 400 μm thick sapphire wafer for microwave delivery (Supplementary Fig. 1). Specifically, photoresist layers (LOR 1A/S1805) are first applied to the sapphire wafer at a spin rate 4000 rpm for 1 min. A carefully designed waveguide pattern with a 50 μm wide central line is then developed using a direct-write lithography system (Heidelburg DWL66+), followed by an $O_2$ plasma cleaning process to remove resist residue. A 10 nm chromium adhesion layer and a 160 nm gold layer are deposited using thermal evaporation, followed by a lift-off process.

## Comparison of $V_B^-$ in hBN with different isotopes

**Simulation of ESR spectrum.** To quantitatively capture the measured ESR spectra in hBN samples with different isotopes, we numerically simulate the $V_B^-$ transitions by accounting for its hyperfine couplings to the nearest 36 nuclear spins. At small external magnetic fields, the hyperfine terms from main text Eq. (1), can be approximated as $\sum_j \mathbf{S}\mathbf{A}^j\mathbf{I}^j \approx \sum_j A_{zz}^j S_z I_z^j = (\sum_j A_{zz}^j I_z^j)S_z$, which leads to an effective energy shift of the electronic spin state depending on the nearby nuclear spin configurations. Considering the hyperfine interaction between $V_B^-$ and an individual nuclear spin with spin number $\mathcal{I}$, the averaged spectrum for the electronic spin transition $|m_s = 0\rangle \leftrightarrow |m_s = \pm 1\rangle$ will exhibit $2\mathcal{I} + 1$ resonances with splitting $A_{zz}$ assuming the nuclear spin is in a fully mixed state (to be expected at room temperature). We note that the value of the hyperfine coupling is proportional to the gyromagnetic ratio of the nuclear spin, i.e. $A_{zz} \propto \gamma_n$; therefore, the overall bandwidth of the transition is proportional to $2\mathcal{I} \times |\gamma_n|$. Supplementary Fig. 2 shows a schematic ESR spectrum assuming the $V_B^-$ couples to a single $^{14}N$ or $^{15}N$ nuclear spin. To achieve a smaller overall linewidth of the $V_B^-$ transitions, the value $2|\gamma_n \mathcal{I}|$ must be minimized for each atomic species in the host material (Fig. 1a).

Following this approximation, we proceed to compute the $V_B^-$ ESR spectrum by summing over the hyperfine couplings to the 18 nearest nitrogen and 18 nearest boron nuclear spins. The values of $A_{zz}$ for $^{14}N$ and $^{11}B$ have been previously predicted using a Density Functional Theory (DFT) calculation[36]. To calculate $A_{zz}$ for $^{15}N$ and $^{10}B$ nuclear spins, we only need to account for the differences between gyromagnetic ratios. Direct summation of all nuclear configurations yields $(2\mathcal{I}_N + 1)^{18} \times (2\mathcal{I}_B + 1)^{18}$ individual resonances, where $\mathcal{I}_N$ and $\mathcal{I}_B$ represent the nuclear spin numbers of the corresponding nitrogen and boron isotopes, respectively. To mitigate the computational cost, we employ Fourier transformation to simplify the calculations (see Supplementary Note 2). The final ESR spectrum is obtained by binning transition frequencies with bin size 0.5 MHz. Supplementary Fig. 3 summarizes the simulated ESR spectra of $V_B^-$ centers in the four different types of isotopically purified hBN crystals ($h^{10}B^{14}N$, $h^{10}B^{15}N$, $h^{11}B^{14}N$, $h^{11}B^{15}N$), as well as naturally abundant hBN. As predicted, $V_B^-$ centers in $h^{10}B^{15}N$ has the least crowded spectrum with the narrowest individual transitions.

**Coherence measurement of $V_B^-$ in other isotopically purified hBN.** To further verify that $h^{10}B^{15}N$ is the ideal host regarding the $V_B^-$ coherence timescales, we prepare another sample with $h^{11}B^{15}N$, which is predicted to be the second best host material. Unsurprisingly, the ESR spectrum of $V_B^-$ in $h^{11}B^{15}N$ exhibits a slightly broader transition linewidth in comparison with $h^{10}B^{15}N$ (Supplementary Fig. 13a). By measuring the Spin Echo coherence time, $V_B^-$ in $h^{11}B^{15}N$ indeed shows $T_2^E = 119 \pm 6$ ns, in between the values measured from $h^{10}B^{15}N$ and $h^{10}B^{15}N$ (Supplementary Fig. S13b). A previous work[28] has also shown that $V_B^-$ in $h^{10}B^{14}N$ and $h^{11}B^{14}N$ did not exhibit a clear improvement in spin echo $T_2$ compared to naturally abundant hBN. Therefore, we believe that $h^{10}B^{15}N$ is indeed the optimal isotope selection for improving the coherence time of $V_B^-$.

## Quantifying nuclear polarization

To quantify the nuclear spin polarization of the nearest three $^{15}N$, we assume each nuclear spin is individually prepared to $|m_I = \uparrow\rangle$ with

probability $P(|\uparrow\rangle)$ and $|m_I = \downarrow\rangle$ with probability $1 - P(|\uparrow\rangle)$. The energy sublevels accounting for the three $^{15}N$ spins should effectively follow a binomial distribution with $n = 3$, the total number of a sequence of independent events. For a random variable $X$ defined as the number of $^{15}N$ nuclear spins being measured to be $|m_I = \uparrow\rangle$ in a single observation, the probability mass function is

$$\Pr(X = k) = \binom{n}{k}P^k(1 - P)^{3-k}. \qquad (6)$$

For example, the four possible nuclear spin configurations correspond to $\sum m_I = -\frac{3}{2}, -\frac{1}{2}, \frac{1}{2}, \frac{3}{2}$ and $X = 0, 1, 2, 3$. Therefore, the theoretical probability distribution follows $\left\{ \binom{3}{3}(1 - P)^3 : \binom{3}{2}P(1 - P)^2 : \binom{3}{1} \right.$ $\left. P^2(1 - P) : \binom{3}{0}P^3 \right\} = \{(1 - P)^3 : 3P(1 - P)^2 : 3P^2(1 - P) : P^3\}$. When there is no nuclear spin polarization ($P = 0.5$), we obtain amplitudes {1 : 3 : 3 : 1} for the ESR spectra.

To quantitatively extract the nuclear polarization, we first fit the ESR spectrum using the sum of four equally spaced Lorentzian distribution whose amplitude follows the aforementioned ratio. However, we note that the gyromagnetic ratio for $^{15}N$ is negative, which reverses the order of the sublevels in the ESR spectrum. In this case, for the $V_B^-$ transition between electronic $|m_s = 0\rangle$ and $|m_s = +1\rangle$ states, the lowest-frequency resonance corresponds to $\sum m_I = \frac{3}{2}$ or $X = 3$ (main text Fig. 3b).

We note here that previous studies have predicted a reduction of symmetry in the $V_B^-$ excited state manifold from $D_{3h}$ to $C_{2v}$ due to Jahn–Teller effect[35,36,64]. This can lead to a different hyperfine interaction for one of the $^{15}N$ nuclear spins compared to the other two. To this end, we also investigate the fitting of the experimental spectra assuming two of the $^{15}N$ nuclear spins are polarized to $|m_I = \uparrow\rangle$ with probability $P_1$ and the third $^{15}N$ is polarized with probability $P_2$. The corresponding probability distribution for the four hyperfine resonances takes the form $\{(1 - P_1)^2(1 - P_2) : (1 - P_1)^2 P_2 + 2P_1(1 - P_1)(1 - P_2) : P_1^2(1 - P_2) + 2P_1 P_2(1 - P_1) : P_1^2 P_2\}$. Supplementary Fig. 4 illustrates a comparison between the two different fitting methods, and we find that both models capture the experimental spectra almost identically well. Thus, the addition of the parameter $P_2$ does not significantly enhance agreement with experiment. Furthermore, the two-parameter fit has much greater uncertainty in the extracted values of $P_1$ and $P_2$ compared to the single-parameter fit and the net nuclear spin polarization amount remains almost exactly the same between the two cases. Therefore, we utilize the first model with minimal fit parameters throughout the manuscript. Further studies are required to experimentally distinguish whether a parameter such as $P_2$ is required.

## Error analysis

The error bars for the measured fluorescence and contrast represent one standard deviation of the statistical error accounting for the photoluminescent counts from experiment. The errors on the measured photoluminescent counts are directly estimated assuming a Gaussian distribution with a mean value $N$ and a standard deviation $\sqrt{N}$, where $N$ is the detected photon number. The error bars for the measured spin coherence time ($T_2$), relaxation time ($T_1$) and nuclear spin polarization represent one standard deviation from the fitting. We note that in our figures, some error bars are smaller than the data markers.

## Data availability

Further data are available from the corresponding author upon request. Source data are provided with this paper.

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

## Acknowledgements

We gratefully acknowledge Vincent Jacques, Cassabois Guillaume, Bernard Gil, Cong Su, Emily Davis, Weijie Wu, Wonjae Lee, Khanh Pham, Benchen Huang, Xingrui Song, Li Yang, Du Li, Xingyu Gao, and Joonhee Choi for helpful discussions. We thank Justin S. Kim, Zhihao Xu, and Sang-Hoon Bae for their assistance in experiment. This work is supported by the Startup Fund, the Center for Quantum Leaps, the Institute of Materials Science and Engineering, and the OVCR Seed Grant from Washington University. T.L. acknowledges support from the Gordon and Betty Moore Foundation grant 10.37807/gbmf12259. V.L., B.Y., and N.Y.Y. acknowledge support from the U.S. Department of Energy through BES grant no. DE-SC0019241 and through the DOE Office of Science, Office of Basic Energy Sciences, Materials Sciences and Engineering Division and the Division of Chemical Sciences, Geosciences and Biosciences at LBNL under Contract no. DE-AC02-05-CH11231. Support of E. Janzen and J. H. Edgar for hBN crystal growth is provided by the Office of Naval Research, award number N00014-22-1-2582. X.D. and E.A.H. acknowledge support from the Moore Foundation Experimental Physics Investigators Initiative award no. 11560. *Note added*: During the completion of this work, we became aware of complementary work studying boron-vacancy centers in both nitrogen and boron isotopes purified hBN crystals (Vincent Jacques, private communication)[65].

## Author contributions

C.Z. conceived the idea. R.G., X.D., Z.L. and G.H. performed the experiment. R.G., X.D., V.L., Z.L., B.Y., N.Y. and C.Z. developed the theoretical models and performed the numerical simulations. R.G., X.D., V.L., Z.L. and C.Z. performed the data analysis. E.J. and J.H.E. grew the hBN samples. R.G., X.D., T.L. and E.A.H. fabricated the microwave delivery system. E.A.H. and C.Z. supervised the project. R.G., X.D., V.L. and C.Z. wrote the manuscript with inputs from all authors.

## Competing interests

The authors declare no competing interests.
