## [Peer Review File · Nature Communications]

REVIEWER COMMENTS

Reviewer #1 (Remarks to the Author):

This manuscript by R. Gong et al describes experimental results in which measurements are performed on boron vacancy (VB-) center defects in flakes of isotopically enriched hexagonal boron nitride (hBN.) The primary results are measurements of narrower spin transition spectra, somewhat extended coherence times, somewhat longer spin T1 lifetimes at low temperatures, and dynamic polarization of nearby nitrogen-15 (^{15}N) nuclear spins in the isotopically enriched samples.

These results are of interest to experts in the emerging field of solid-state defects in 2D materials. The results are also clearly presented (for the most part, see my specific comments below). However, in my opinion the demonstrated gains in coherence and the achieved degree of nuclear spin polarization and control are fairly modest and are not nearly large enough to impact whether or not these defects can be used for any of the applications the authors list (quantum sensing, quantum registers). In addition, the fact that all the measurements were performed with ensembles of defects in flakes many layers thick (I believe, please see my specific comments below) also reduces their impact and relevance to these applications. I therefore feel these results are more appropriate for a more field specific journal such as Communications Physics, and do not recommend publication in Nature Communications. I also have a number of specific comments and questions that I feel the authors should address before this manuscript is published in any journal.

Specific comments and questions:

1) The authors predict that $h^{10}\text{B}^{15}\text{N}$ is the optimal host material, but I don't feel they convincingly demonstrate this. They demonstrate that isotopic purification provides somewhat better coherence times, which makes sense because it reduces inhomogeneity between the defects, but it's not at all clear to me if this is because ^{10}B and ^{15}N are the optimal isotopes, or if it is just better to have all nearby nuclear spins of each element be of the same isotopic species, regardless of that species. Is it possible that the authors' logic is wrong or they have overlooked something and VB- centers in $h^{11}\text{B}^{15}\text{N}$, $h^{11}\text{B}^{14}\text{N}$, or $h^{10}\text{B}^{14}\text{N}$ would have equal or even longer coherence times to those in $h^{10}\text{B}^{15}\text{N}$?

2) How isotopically pure are the samples? Possibly I missed it, but I think the only place where this is discussed at all is in the "Methods" section where it says the flakes are grown with "> 99%" ^{10}B and ^{15}N , but this is not very specific. After all, naturally abundant hBN is already 99.6% ^{14}N .

3) The DC and AC magnetic field sensitivities quoted in this work seem particular to these samples, in that they depend on how many defects contribute to the measurements and on the collection efficiency of the setup used, etc. How dense are the defects in the samples? How many defects contribute to these measurements?

4) Related to question 3, how many layers do the authors believe are in the flakes used for these measurements? The only information on this I found was one mention in the "Methods" that the flakes are roughly 100 nm thick, but possibly I missed more detailed discussion of this?

5) Do the authors have any explanation for why the T1 times of VB- defects in the isotopically enriched samples are longer than those of the VB- defects in the naturally abundant samples at lower temperatures, but not at room temperature? What are the physical mechanisms that are limiting T1 in these regimes?

6) The authors' write that the fit in Fig 4C "...corresponds to a cosine fit with exponentially decaying amplitude." But the behavior of the fit at very short times does not resemble a decaying cosine at all. What is the functional form of this fit? What is happening at short times? In addition, the data seems to exhibit beating that is not captured by the fit. Where is this coming from? Can this behavior captured by a slightly more sophisticated model that accounts for some inhomogeneity in the three N15 nuclear spins?

Reviewer #2 (Remarks to the Author):

Spin defects in wide bandgap semiconductors are regarded as promising candidates for quantum technologies such as spin qubits and quantum sensors working at room temperature. Especially, since spin defects in 2D materials can be applied to flexible applications, the topic of this manuscript is suitable for nature communications, I think. The isotope engineering is an important way to improve spin properties. Actually, nitrogen-vacancy in diamond as well as silicon vacancy in SiC can achieve longer spin coherence time by isotope engineering. For hBN, this study firstly showed the improvement of spin properties for spin defect (Vb) in hBN by isotope engineering. In principle, isotope engineering works for any spin defects. However, in reality, it is important to observe such improvement by experiments. The results shown in the manuscript is well understood because authors obtained those results by careful/accurate measurements. However, I would suggest/confirm followings before accept for publication.

[1] [last part of the introduction] Authors mention the DC magnetic field sensitivity “DC magnetic field sensitivity $\eta_{DC} \approx 10 \mu\text{T Hz}^{-1}$ but That for AC $\eta_{AC} \approx 46 \mu\text{T Hz}^{-1}$ ”. Is it correct? Usually, AC sensitivity can extend by XY8 but not for DC?

[2] Authors describe measurement procedures using CFM as “Supplementary Information”. However, this information is very important for authors to understand the experiments done by authors. So, I would suggest that authors describe the measurement procedures using CFM in the main text.

[3] For the symbols in Fig. 2, only circle is used. It is a bit difficult for readers to understand the result. Especially, for Fig. 2 (c), both results obtained from h10B15N and hBNnat in the (almost) same color (green). So, please use different symbols such as square or triangle for hBNnat.

[4] [In the “Experimental characterization” section, “B=90G” is described in the text to explain about the result in Fig.1 (c). however, in the figure caption, the value of the magnetic field is written to be “87G”. Please correct the value of the magnetic field.

[5] [minor but important] For the numbering of affiliation, “6” is used two times but missing “5”. Please correct.

Takeshi Ohshima

Reviewer #3 (Remarks to the Author):

Gong et. al. report on enhanced spin properties of the negatively charged boron vacancy in hBN utilizing isotopically pure hBN crystals for the first time. The authors demonstrate distribution and coherence of VB- spin transitions, improved performance as a magnetic field sensor, coherent control/readout of nuclear coupling, and identify discrepancies with ab-initio calculations in the existing literature. It is overall a nice and noteworthy work, with aspects to appeal to a reasonably broad audience. The methodology, experiments, and data presentation are well done. However, the writing and organization of the manuscript requires substantive changes to improve clarity and comprehension for publication. Some specific points on this below, as well as a few technical questions.

- 1) Given it's a new hBN material type for studying these defects, and there is significant variability between different hBN materials and the resulting defects properties, some basic optical characterization would be useful to include. In the SI is fine, but spectra at temp used in most experiments, optical saturation measurement (does it match the value for the spin polarization saturation measurement)? Count rates observed giving Vb defect density in this experiment.
- 2) I don't think the temperature of the measurements is defined anywhere other than specifically in the temperature dependent figure 2c. What is temp for rest of data?
- 3) "Here, we find the steepest slopes are $8.2 \times 10^{-11} \text{ Hz}^{-1}$ and $3.0 \times 10^{-10} \text{ Hz}^{-1}$ for hBNnat and h10B15N respectively, and after accounting their similar photon detection rate and ESR contrast, the sensitivity enhancement increases to ~ 4 -fold." Not clear to me if there was some correction applied due to the photon detection or ESR contrast? If so should mention this briefly in the main text.
- 4) Discussion on the discrepancy of ab-initio works with the results should be discussed in further detail in the main text. Are there other relevant experimental works with discrepancy with the ab-initio works? What future experiments may help address this?
- 5) Unclear if the simulated fit for the 210G nuclear spin resonance measurement in Fig S9 is fit from the Ab-initio or the adjusted values. It's clear the fit at 760G is different from the one in figure 4b. If it is the case that the ab-initio fits similar to the adjusted at lower fields this should be mentioned much more clearly.
- 6) "First, for sensing applications, we believe the V-B defects in our isotopically purified host h10B15N outperform conventional naturally abundant hBNnat in all aspects." Would be more appropriate to specify aspects you demonstrated improvement. Are you also suggesting enhanced pressure sensing, if so why?
- 7) There are significant portions of the manuscript which would greatly benefit from a bit of rewriting/reorganizing for clarity.
 - a) Mixing of experimental details/results/theory/interpretation can be hard to follow in some places. Take for instance the discussion of the T1 improvement. This is attributed to the heavier N15 mass twice before ever presenting the results. But the only mention of whether this fits the acquired data that both hBN types increase monotonically in T1 with decreasing temperature, and not if the heavier N15 mass fits the observed increase in T1 specifically. The isotopically engineered hBN also has an average lighter B mass, is this insignificant compared to the nearest neighbor N15's?
 - b) Another example, the introduction section reads a bit more like a conclusion than any conclusion portion.
 - c) In general, the relevant discussions are included, but at times scattered in many places, or experimental details discussed multiple paragraphs before discussion of the experiments results.
- 8) Ref #4 Listed twice in first citations
- 9) Typo SI, 1.2 section title "defects"

We sincerely thank all the referees for their tremendously valuable comments, questions, and suggestions. In particular, we are grateful for Referee 2 and Referee 3 being interested in our result and supportive of the publication, commenting that “(Referee 2) the topic of this manuscript is suitable for nature communication, ..., the results shown in the manuscript is well understood because authors obtained those results by careful/accurate measurements”, and “(Referee 3) it is overall a nice and noteworthy work, with aspects to appeal to a reasonably broad audience. The methodology, experiments, and data presentation are well done.” We also thank Referee 1 for their positive comment “(Referee 1) These results are of interest to experts in the emerging field of solid-state defects in 2D materials.”.

However, the referees also bring up a number of very important questions regarding the potential applications and prospects of the V_B^- defects, experimental details and underlying physical implications, as well as the organization of the manuscript. We totally agree with the referees and apologize for not being clear and precise in the original manuscript. We have carefully re-analyzed the experimental results, as well as performing some additional measurements to justify our observation more clearly. Finally, we have tried our very best to carefully respond to all of the referee’s suggestions; this has led to some extensive rewriting of the manuscript. We truly believe that the manuscript we now resubmit is substantially improved thanks to all the referees’ comments and hope the referees will now support the publication in Nature Communications.

Reviewers Comments:

Reviewer #1 (Remarks to the Author):

This manuscript by R. Gong et al describes experimental results in which measurements are performed on boron vacancy (VB-) center defects in flakes of isotopically enriched hexagonal boron nitride (hBN.) The primary results are measurements of narrower spin transition spectra, somewhat extended coherence times, somewhat longer spin T1 lifetimes at low temperatures, and dynamic polarization of nearby nitrogen-15 (^{15}N) nuclear spins in the isotopically enriched samples.

These results are of interest to experts in the emerging field of solid-state defects in 2D

materials. The results are also clearly presented (for the most part, see my specific comments below). However, in my opinion the demonstrated gains in coherence and the achieved degree of nuclear spin polarization and control are fairly modest and are not nearly large enough to impact whether or not these defects can be used for any of the applications the authors list (quantum sensing, quantum registers). In addition, the fact that all the measurements were performed with ensembles of defects in flakes many layers thick (I believe, please see my specific comments below) also reduces their impact and relevance to these applications. I therefore feel these results are more appropriate for a more field specific journal such as Communications Physics, and do not recommend publication in Nature Communications. I also have a number of specific comments and questions that I feel the authors should address before this manuscript is published in any journal.

We must thank the referee for the thorough review of our manuscript, and we are also grateful for the valuable comments and questions, which we have tried our best to address below. At the same time, we understand that the referee has reservations about the significance of our result and would like to take this opportunity to plead our case by contextualizing our findings in a broader context.

The recent emergence of the search for spin defects in atomically-thin materials is motivated by the possibility of many quantum applications with dimensional advantages when compared to their 3D counterparts, such as NV centers in diamond. In particular, from the perspective of quantum sensing, the atomically-thin layered structure of the host materials allows the sensors to be easily positioned in close proximity with the target materials. Moreover, since hBN has already been widely employed as the encapsulation and gating dielectric material in 2D heterostructure devices, embedding V_B^- as sensors to investigate 2D materials does not involve any additional complexity in the fabrication process. As a result, although the current ESR linewidth and coherence time (thus the sensitivity) of V_B^- are worse than the values reported in NV centers, V_B^- has still been proven to be a powerful and unique quantum sensor for imaging 2D magnetism [1–4]. Besides magnetic field sensing, the measured temperature and strain response of V_B^- spin state is around 20 times better than NV

centers in diamond [1, 5], originating from the relatively soft lattice structure of the hBN crystal. This attribute of V_B^- makes it a promising candidate for sensing the local changes in temperature and strain [1, 6, 7].

In our work, we demonstrate that isotope engineering is a fundamental and universal strategy for improving the spin properties and thus the sensitivity of the 2D spin defects. Isotope engineering is universal in the sense that *almost all* previous works using V_B^- for sensing applications could greatly benefit from simply switching to our newly prepared $h^{10}B^{15}N$ sample! This sensitivity improvement is also fully compatible with other ongoing V_B^- optimization efforts, such as dynamical decoupling protocols to extend T_2 [8, 9], coupling V_B^- to optical cavities [10–12] and increasing sensor number and density [9, 13, 14] to improve fluorescent signals.

From the perspective of nuclear spin control and quantum register, we agree with the referee that the current nuclear spin polarization level ($\lesssim 30\%$) is still quite moderate. However, compared to prior studies [15] using natural abundant samples, our isotopically purified $h^{10}B^{15}N$ exhibits well-resolved hyperfine levels which enables a nuclear spin polarization characterization that is the most accurate to date. Moreover, our work identifies the main cause that limits the current nuclear spin polarization level, i.e. the $[A_2^j S_+ I_+ + h.c.]$ term in the hyperfine interaction Hamiltonian. These advances, together with the observed individually addressable hyperfine levels and sharp nuclear spin transitions in $h^{10}B^{15}N$ sample, will help us design experimental protocols to further improve the nuclear spin polarization level in future.

In addition, the referee is correct that all our measurements on V_B^- are performed with ensembles of defects in multi-layered hBN flakes (ranging from 20 – 70 nm thick, see our response to the second question and Supplementary Note 8). In fact, the relatively weak fluorescent signals of V_B^- prevents the isolation of single spin defect in all prior works. Therefore, we totally agree with the referee that compared to single NV sensing experiment, at the moment ensemble of V_B^- centers (or NV centers) cannot reach sub-diffraction limit spatial resolution. However, we note that, even for ensemble sensing applications, the V_B^- ensemble, as well as NV ensemble, have been demonstrated as a powerful and

useful toolset to image magnetism, stress and temperature with sub-micrometer resolution [1–4, 6, 7, 13]. Moreover, thanks to the simple lattice structure of V_B^- which only involves a lattice vacancy, one can create a much higher sensor density ($\gtrsim 300$ ppm) [9] compared to NV ensemble in diamond (~ 5 ppm). Such high sensor density partially compensate the low fluorescent signals of individual V_B^- center. As a result, when comparing fluorescent signals, V_B^- ensemble exhibits similar fluorescent count level with NV ensemble for a same defect layer thickness.

To summarize, we believe that our work has demonstrated isotope engineering as a foundational method to improve the quantum properties of spin defects in 2D materials, which represents a significant progress in this blooming field. As suggested by the other two referees, “(Referee 2) since spin defects in 2D materials can be applied to flexible applications, the topic of this manuscript is suitable for nature communication”, and “(Referee 3) It is overall a nice and noteworthy work, with aspects to appeal to a reasonably broad audience.” To better emphasize the significance and broad impact of our work, we have made substantial changes in the introduction and conclusion sections of the revised manuscript. We sincerely hope that with these changes and our point-by-point response below, the referee now will support the publication of our work in **Nature Communications**.

Specific comments and questions:

1) The authors predict that h10B15N is the optimal host material, but I don’t feel they convincingly demonstrate this. They demonstrate that isotopic purification provides somewhat better coherence times, which makes sense because it reduces inhomogeneity between the defects, but it’s not at all clear to me if this is because 10B and 15N are the optimal isotopes, or if it is just better to have all nearby nuclear spins of each element be of the same isotopic species, regardless of that species. Is it possible that the authors’ logic is wrong or they have overlooked something and VB- centers in h11B15N, h11B14N, or h10B14N would have equal or even longer coherence times to those in h10B15N?

We thank the referee for bringing up a very good question and apologize for not being clear enough in our original manuscript. To answer this question, we

need to first determine what limits the coherent timescales of V_{B}^- in our experiment, and the main decoherence source is the hyperfine interaction to nearby N and B nuclear spins, as represented by the term: $\sum_j SA^j I^j$. Here we would like to briefly reiterate our reasoning in the section “Optimal Isotopes of hBN”. We first compare the effect of all four different possible combinations of isotopes and determine that $h^{10}\text{B}^{15}\text{N}$ is the best host material that gives the narrowest ESR transition linewidth after accounting for both the nuclear spin degree of freedoms and gyromagnetic ratios. To further validate this, we numerically simulate the V_{B}^- transitions by accounting for its hyperfine couplings to the nearest 36 nuclear spins (Methods Section 2: Simulation of ESR spectrum in hBN with different isotopes). Supplementary Figure S3 summarizes the simulated ESR spectra of V_{B}^- centers in the four different types of isotopically purified hBN crystals, where, as predicted, V_{B}^- in $h^{10}\text{B}^{15}\text{N}$ exhibits the least crowded spectrum with the narrowest individual transitions. We note that, in a complementary work that is submitted together with our result, Clua-Provost et al. [16] has reached to the same conclusion. In particular, they experimentally measure the ESR spectra of V_{B}^- in $h^{10}\text{B}^{14}\text{N}$, $h^{10}\text{B}^{15}\text{N}$, and $h^{11}\text{B}^{15}\text{N}$. As expected, they found that $h^{10}\text{B}^{15}\text{N}$ is indeed the optimal hosting material regarding the narrowest V_{B}^- transition linewidth.

Narrower transition linewidth typically corresponds to longer coherence time, T_2 . Our work carries out the first experimental investigation of the coherence time in $h^{10}\text{B}^{15}\text{N}$. Compared to natural abundant $h\text{BN}_{\text{nat}}$, we achieve an approximate two-fold improvement in both spin echo and XY8 T_2 timescales. To address referee’s concern about coherence timescales in other isotopically purified hBN crystal, we prepare another sample with $h^{11}\text{B}^{15}\text{N}$, which is predicted to be the second best host material. By measuring the Spin Echo coherence time on that flake, we can clearly see that, V_{B}^- in $h^{10}\text{B}^{15}\text{N}$ indeed shows the longest T_2 , and $h^{11}\text{B}^{15}\text{N}$ gives the second longest T_2 , while natural abundant hBN gives the shortest (see Figure R1).

At the moment, we do not have access to the other two isotopically purified crystals with ^{14}N nuclei. However, a previous work [17] has shown that V_{B}^- in $h^{10}\text{B}^{14}\text{N}$ and $h^{11}\text{B}^{14}\text{N}$ did not exhibit a clear improvement in spin echo T_2 compared

FIG. R1: $h^{11}B^{15}N$ **characterization**. Added Figure S13 from Supplementary Information. (a) The ESR spectra of V_B^- in $h^{10}B^{15}N$ and $h^{11}B^{15}N$. (b) The spin echo measurement.

to natural abundant hBN. Combining all these together, we believe that $h^{10}B^{15}N$ is indeed the optimal isotope selection for improving the coherence time of V_B^- , which agrees with the theoretical prediction from both our work and the complementary work [16].

To this end, in the revised manuscript, we have included our new T_2 measurement on the $h^{11}B^{15}N$ sample into the Methods part for a better comparison. We thank the referee again for this question that helps us better contextualize our result.

2) How isotopically pure are the samples? Possibly I missed it, but I think the only place where this is discussed at all is in the “Methods” section where it says the flakes are grown with “> 99%” ^{10}B and ^{15}N , but this is not very specific. After all, naturally abundant hBN is already 99.6% ^{14}N .

We apologize for not being more specific in our original manuscript. The hBN crystals that we investigate in this work were grown by the metal flux method using boron and nitrogen single isotope enriched sources [18] with purity $^{10}B > 99.2\%$ and $^{15}N > 99.7\%$. Therefore, the final compound should also have a similar concentration. We agree with the referee that the natural abundant concentration for nitrogen is already 99.6% of ^{14}N ; but in our work, one of the main contributions is that we, for the first time, purify the nitrogen isotope in hBN to be the non-abundant ^{15}N , which leads to substantial improvements of

the V_B^- spin properties.

To this end, in the revised manuscript, we have also included the precise values of the isotope purity into the main text to better present our result. We thank the referee again for bringing up this question.

3) The DC and AC magnetic field sensitivities quoted in this work seem particular to these samples, in that they depend on how many defects contribute to the measurements and on the collection efficiency of the setup used, etc. How dense are the defects in the samples? How many defects contribute to these measurements?

We completely agree with the referee that DC and AC magnetic field sensitivities depend on a myriad of factors including the collection efficiency and the number of V_B^- contributing to measurements. As shown in Equation R1,

$$\eta_{\text{DC}} \approx \frac{2\pi}{\gamma_e \sqrt{R}} \left(\max \left| \frac{\partial C(\nu)}{\partial \nu} \right| \right)^{-1}, \quad \eta_{\text{AC}} \approx \frac{\pi}{2\gamma_e C_{\text{max}} e^{-1} \sqrt{n_{\text{avg}}}} \frac{\sqrt{t_I + \tau + t_R}}{\tau} \quad (\text{R1})$$

we can see that both the photon detection rate, R , and the signal contrast, C , are hBN sample and set-up dependent. From our previous study of V_B^- densities created via the same method, the density is estimated to be ≈ 150 ppm [9]. With diffraction-limited laser spot of diameter $\sim 0.6 \mu\text{m}$ and the created defect thickness ~ 60 nm (which is determined by the ion implantation energy), we estimate the number of defects contributed to this measurement is around 2.6×10^5 .

Indeed, as the referee correctly points out, if one increases the size of the laser spot and the thickness of defects created, we can increase the defect number and the total photon detection rate (R), thus improving the sensitivity. However we note that this will also sacrifice the spatial resolution of the sensors. Indeed, there are several recent work that trying to optimize the sensitivity by using a much larger ion implantation energy and dosage, as well as thicker hBN samples [13, 14].

In our revised manuscript, we add a short discussion in the main text to better clarify that our sensitivity estimation is sample specific. We thank the referee again for bringing this up that helps to improve our manuscript.

4) Related to question 3, how many layers do the authors believe are in the flakes used for these measurements? The only information on this I found was one mention in the "Methods" that the flakes are roughly 100 nm thick, but possibly I missed more detailed discussion of this?

We appreciate this question from the referee which has also helped us better refine our results. Originally, we estimate the thickness of our samples by identifying the color of flakes under our microscope, which is a common empirical technique in 2D materials to estimate the flake thickness (Fig. R2a). To address the referee's question and quantitatively characterize the thickness of the hBN flakes, we have performed additional thickness measurement of hBN flakes using atomic force microscopy (AFM). In particular, we conducted AFM on ~ 30 separate hBN flakes with varying thicknesses, establishing a reliable mapping from the color of the hBN flakes under our microscope to the precise thickness (Fig. R2b). From the figure, we calibrate the thickness of the six flakes studied in our manuscript and summarize them in Table R1.

FIG. R2: **Thickness calibration curve** Added Figure S12 in Supplementary Information. (a) A common optical microscopy image of varying thickness hBN flakes on Si wafer with 300 nm thermal oxide. (b) Plotting of AFM thickness and wavelength estimation from optical image. The dashed line shows the second degree polynomial fitting. The shadowed region indicates the standard derivation of $\sigma \sim 8$ nm. The second degree polynomial fitting shown in the figure was then used for estimating the sample thickness in this paper.

In the revised manuscript, we have added a new section in the supplemental information for a detailed discussion of the thickness characterization process. We thank the referee for this nice question that helps us better present our

	S1	S2	S3	S4	S5	S6
Thickness (nm)	64 ± 8	63 ± 8	63 ± 8	72 ± 8	70 ± 8	21 ± 8

TABLE R1: Summary of the estimated flake thickness for the six hBN samples presented in main text Figure 2.

results.

5) Do the authors have any explanation for why the T_1 times of V_B^- defects in the isotopically enriched samples are longer than those of the V_B^- defects in the naturally abundant samples at lower temperatures, but not at room temperature? What are the physical mechanisms that are limiting T_1 in these regimes?

We sincerely thank the referee for highlighting this, and we have to admit that the spin relaxation mechanism for V_B^- is still not fully understood and invites further studies. In fact, the spin relaxation mechanism is still an actively studied subject even for well-established spin defect systems like the NV center in diamonds [19].

Specifically, we first note that, the measured increasing trend of T_1 with lowering temperatures suggests that the V_B^- spin relaxation is mainly limited by the coupling between the V_B^- spin state and the phonon motion inside hBN. The temperature dependence of the V_B^- spin-phonon interaction has yet to be systematically investigated. Intuitively, when the atom mass increases, the spin-phonon coupling strength is expected to decrease; however the magnitude of changes could highly depend on if the spin-phonon coupling is a first-order or higher-order process. At different processes, the mass dependence can be very distinct, as suggested by this ab-initio study [20]. In this case, one potential explanation for why the difference of T_1 is only observed at lower temperatures is that at different temperature ranges, the dominant spin-phonon coupling term could have different mass dependent relationship. We have actually just started branching out to systematically understand the underlying mechanism of V_B^- spin-phonon relaxation process from ab-initio calculations, but the progress is

still at very early stages.

We have to admit that currently we don't have a very quantitative explanation for temperature and mass dependence of T_1 in V_B^- . To this end, following the suggestion from Referee 3, in the revised manuscript, we switch the order of the associated discussion to mainly focus on the experimental observations, and leave the underlying physical mechanism as an open question for future studies. We thank the referee again for this question that help us better elucidate our observation and understanding.

6) The authors' write that the fit in Fig 4C "... corresponds to a cosine fit with exponentially decaying amplitude." But the behavior of the fit at very short times does not resemble a decaying cosine at all. What is the functional form of this fit? What is happening at short times? In addition, the data seems to exhibit beating that is not captured by the fit. Where is this coming from? Can this behavior captured by a slightly more sophisticated model that accounts for some inhomogeneity in the three N15 nuclear spins?

We thank the referee for pointing out this potential confusion and apologize for not providing more detailed discussions in the original manuscript. The cosine fit with exponentially decaying amplitude takes the form $Ae^{-(t/T)^\alpha} \cdot \cos(2\pi t/\Omega) + c$, which is the most straightforward form of fitting if one assumes each nuclear spin is isolated. However, as the referee correctly points out, in reality, the three nuclear spins are effectively interacting with each other (intermediated via the hyperfine interaction with V_B^- electronic spin), which requires a more sophisticated model to capture the underlying dynamics.

In our Supplementary Information Note 6.3 and Figure S10, we generate and diagonalize the system Hamiltonian including the V_B^- electronic spin ground state and the three nearest-neighbour ^{15}N nuclear spins, and obtain the associated eigenstates v and eigenenergy E . The RF field we use in the nuclear spin Rabi measurement drives transition from an initial state v_i to a final state v_f that has an energy $\Delta E = |E_f - E_i|$ and amplitude $T_{i \rightarrow f} = |\langle v_f | S_x | v_i \rangle| + |\langle v_f | S_y | v_i \rangle|$. Summing up all the simulated nuclear Rabi signals corresponding to the many possible transitions leads to the "initial drop" and "beating" that qualitatively match our experimental observations (Figure R3, which is also attached here). However,

we would also like to emphasize that the simulated dynamics is highly sensitive to the transverse hyperfine coefficients of the nuclear spins, which have not been accurately measured from experiment. Currently we rely on the rescaled values calculated from ab-initio, which could lead to the mismatch between simulation and experiment. Moreover, if one directly uses the ab-initio calculated values to simulate the nuclear spin Rabi experiment, we obtain a much faster oscillation that does not match the experimental data (Figure R3b).

FIG. R3: **Supplementary Information Figure S10** (a) Simulation of the nuclear spin Rabi measurement using the adjusted transverse hyperfine coefficients. (b) Simulation of the nuclear spin Rabi measurement using the ab-initio calculated transverse hyperfine coefficients.

To this end, we add a short discussion to the main text to emphasize this interacting nature of the three ^{15}N nuclear spins, and explicitly refer the readers to our more sophisticated model in the supplementary materials. We thank the referee for this very important question that helps us better clarify our work.

Reviewer #2 (Remarks to the Author):

Spin defects in wide bandgap semiconductors are regarded as promising candidates for quantum technologies such as spin qubits and quantum sensors working at room temperature. Especially, since spin defects in 2D materials can be applied to flexible applications, the topic of this manuscript is suitable for nature communications, I think. The isotope engineering is an important way to improve spin properties. Actually, nitrogen-vacancy in diamond as well as silicon vacancy in SiC can achieve longer spin coherence time by isotope engineering. For hBN, this study firstly showed the improvement of spin properties for spin defect (Vb) in hBN by isotope engineering. In principle, isotope engineering works for any spin defects. However, in reality, it is important to observe such improvement by experiments. The results shown in the manuscript is well understood because authors obtained those results by careful/accurate measurements. However, I would suggest/confirm followings before accept for publication.

We sincerely thank the referee for the careful reading of our work and the support for publication. We are also very grateful for the constructive suggestions and questions that help us correct and improve our manuscript.

[1] [last part of the introduction] Authors mention the DC magnetic field sensitivity “DC magnetic field sensitivity “ $\eta_{DC} \approx 10 \mu\text{T Hz}^{-\frac{1}{2}}$ but that for AC “ $\eta_{AC} \approx 46 \mu\text{T Hz}^{-\frac{1}{2}}$ ”. Is it correct? Usually, AC sensitivity can extend by XY8 but not for DC?

We must thank the referee for bringing this up. In our initial submission, we mistakenly omit a factor of 2π in the electronic spin gyromagnetic ratio, γ_e , during the calculation of AC magnetic field sensitivity. We have corrected this error in the revised manuscript, and now estimate the $\eta_{AC} \approx 7 \mu\text{T Hz}^{-\frac{1}{2}}$ to be indeed smaller than the DC sensitivity. We thank the referee again for the keen awareness of the scientific validity.

[2] Authors describe measurement procedures using CFM as “Supplementary Information”. However, this information is very important for authors to understand the experiments done by authors. So, I would suggest that authors describe the measurement procedures using CFM in the main text.

We completely agree with the referee that the lack of methodology description in the main text can be confusing. Following referee’s suggestion, we have added a short description of the confocal microscope setup at the end of the first paragraph in “Experimental Characterization section” to guide the audience to Supplemental Information for more detailed measurement procedures.

[3] For the symbols in Fig. 2, only circle is used. It is a bit difficult for readers to understand the result. Especially, for Fig. 2 (c), both results obtained from $h^{10}B^{15}N$ and hBN_{nat} in the (almost) same color (green). So, please use different symbols such as square or triangle for hBN_{nat} .

We appreciate this nice suggestion! We have adopted triangles to denote measurements on $h^{10}B^{15}N$ to better distinguish them from those on hBN_{nat} in Figure 2 as shown below. The updated symbols have indeed improved the readability of the figure.

FIG. R4: Coherent dynamics of V_B^- in isotopically distinct hBN samples Revised Figure. 2 from the main text.

[4] [In the “Experimental characterization” section, “B=90G” is described in the text to explain about the result in Fig.1 (c). however, in the figure caption, the value of the magnetic field is written to be “87G”. Please correct the value of the magnetic field

We thank the referee for identifying this inconsistency. We have verified the value of the magnetic field and corrected the value in the “Experimental Characterization” section to $B \approx 87$ G.

[5] [minor but important] For the numbering of affiliation, “6” is used two times but missing “5”. Please correct. Takeshi Ohshima

We appreciate the referee for pointing out this error and have corrected the relevant numbering.

Again, we are genuinely grateful for referee’s helpful suggestions and the endorsement of our work.

Reviewer #3 (Remarks to the Author):

Gong et. al. report on enhanced spin properties of the negatively charged boron vacancy in hBN utilizing isotopically pure hBN crystals for the first time. The authors demonstrate distribution and coherence of VB⁻ spin transitions, improved performance as a magnetic field sensor, coherent control/readout of nuclear coupling, and identify discrepancies with ab-initio calculations in the existing literature. It is overall a nice and noteworthy work, with aspects to appeal to a reasonably broad audience. The methodology, experiments, and data presentation are well done. However, the writing and organization of the manuscript requires substantive changes to improve clarity and comprehension for publication. Some specific points on this below, as well as a few technical questions.

We are sincerely thankful for the referee's comprehensive review of our manuscript and support of our work. We must also thank the referee for constructive comments on the writing and organization of the manuscript. Following the referee's suggestion, we have made changes throughout the manuscript, which should hopefully improve the overall readability.

1) Given it's a new hBN material type for studying these defects, and there is significant variability between different hBN materials and the resulting defects properties, some basic optical characterization would be useful to include. In the SI is fine, but spectra at temp used in most experiments, optical saturation measurement (does it match the value for the spin polarization saturation measurement)? Count rates observed giving Vb defect density in this experiment.

We appreciate this nice suggestion! Following the suggestions, we have performed additional optical characterizations of V_B^- in $h^{10}B^{15}N$ flakes and include them in the new section of the Supplemental Information. Specifically, we measure the saturation curve of fluorescence against laser power (Fig. R5a) and find that the fluorescence indeed deviate from the linearly increasing and start to saturate around the same laser power (~ 10 mW) that nuclear spin polarization saturates. We also record the optical fluorescence spectra and observe almost no difference between V_B^- in $h^{10}B^{15}N$ and natural abundant hBN_{nat} . The V_B^- density is estimated to be around 150 ppm from the ion implantation dosage and energy

FIG. R5: **Optical characterization of $h^{10}B^{15}N$.** Added Figure S11 from the Supplementary Information. (a) Optical saturation curve of V_B^- in $h^{10}B^{15}N$ against laser power. (b) Photoluminescence spectrum of V_B^- in hBN_{nat} , $h^{10}B^{15}N$, and $h^{11}B^{15}N$.

[9].

To this end, we have added a new section “Optical Characterization of V_B^- in $h^{10}B^{15}N$ ” into the supplementary materials to summarize the results. We thank the referee again for this nice suggestion that help improve our manuscript.

2) I don’t think the temperature of the measurements is defined anywhere other than specifically in the temperature dependent figure 2c. What is temp for rest of data?

We apologize for not clarifying temperature of the measurements in our original manuscript. All measurements other than Figure. 2c are conducted at room temperature $\sim 300K$. We have now added a sentence at the end of the first paragraph in Experimental Characterization section to specify the temperature of measurements and guide the audience to Supplemental Information for a more detailed discussion of measurement procedures.

3) “Here, we find the steepest slopes are $8.2 \times 10^{-11} \text{ Hz}^{-1}$ and $3.0 \times 10^{-10} \text{ Hz}^{-1}$ for hBN_{nat} and $h^{10}B^{15}N$ respectively, and after accounting their similar photon detection rate and ESR contrast, the sensitivity enhancement increases to ~ 4 -fold.” Not clear to me if there was some correction applied due to the photon detection or ESR contrast? If so should mention this briefly in the main text.

We completely agree with the referee that the original wording is not very

clear. In the equation we use to estimate the DC field sensitivity (Eq. R1), both the photon detection rate and ESR contrast contribute to η_{DC} . Therefore, the photon detection rate of hBN_{nat} and $\text{h}^{10}\text{B}^{15}\text{N}$ needs to be similar for a fair comparison of η_{DC} using just $\frac{\partial C(\nu)}{\partial \nu}$. This is indeed the case, and we did not apply any correction in the ~ 4 -fold sensitivity estimation. We have revised this corresponding sentence to clarify that we have similar count rates of V_{B}^- in different isotope samples and are only comparing $\frac{\partial C(\nu)}{\partial \nu}$. We thank the referee for this nice suggestion.

4) Discussion on the discrepancy of ab-initio works with the results should be discussed in further detail in the main text. Are there other relevant experimental works with discrepancy with the ab-initio works? What future experiments may help address this?

We thank the referee for this question which we have also deliberated for a long time ourselves. First of all, while one can straightforwardly obtain the value of A_{zz} from experiment by analyzing the hyperfine splittings from the ESR spectra, other coefficients such as A_{xx} and A_{yy} are much harder to determine in experiments since their associated terms in the spin Hamiltonian are highly suppressed under the secular approximation. To the best of our knowledge, there is only one experimental study that directly investigates the ab-initio calculation of the V_{B}^- transverse hyperfine coefficients for hBN_{nat} with three closest ^{14}N nuclei [15]. In this work, Gao et al. report a $\sim 30\%$ discrepancy between the calculated and experimentally estimated transverse hyperfine coefficients using nuclear spin Rabi measurement. We note that, however, due to the complex nuclear spin structure of ^{14}N , the measured nuclear spin transitions in that work contain multiple resonances and are much broader than what we have seen using ^{15}N nuclei. Therefore we believe that our characterization of the extremely sharp ^{15}N nuclear spin transitions would give a more precise estimation of the transverse hyperfine coefficients.

For the future experiments that may help to better determine the transverse hyperfine coefficients, we envision one can perform a detailed and high resolution pulsed ESR spectroscopy near the ground state anti-crossing (gsLAC), with magnetic field $B \sim 1200$ G. At such external magnetic field, the V_{B}^- electronic

spin state $|0\rangle$ and $|-1\rangle$ are close to degenerate, where the secular approximation no longer holds. In this case, we expect the transverse hyperfine coefficients to significantly affect the shape and the structures of the ESR spectra.

Following the referee's suggestions, we have now added some additional discussions of the observed discrepancy, as well as the potential future experiment into the revised main text. We thank the referee again for this nice suggestion that allow us to better contextualize our result.

5) Unclear if the simulated fit for the 210G nuclear spin resonance measurement in Fig S9 is fit from the Ab-initio or the adjusted values. It's clear the fit at 760G is different from the one in figure 4b. If it is the case that the ab-initio fits similar to the adjusted at lower fields this should be mentioned much more clearly.

The referee makes a very good point and we apologize for not making it clearer that the ab-initio calculated values does fit of the nuclear magnetic resonance at lower field reasonably well. At small magnetic fields, as we mentioned above, the effect of transverse hyperfine coefficients is highly suppressed by the large splitting between $V_B^- |0\rangle$ and $|-1\rangle$ spin states. As a result, the simulation with the calculated and adjusted values fit the experimental data equally well. At large external magnetic field $\sim 760 G$, however, the electronic spin splitting between $|0\rangle$ and $|-1\rangle$ become much smaller, and the effect of transverse hyperfine coefficients becomes evident.

In the revised manuscript, we add a sentence in the caption to clarify that the fit for figure S9 is using the ab-initio values. We also add a short discussion in the Supplementary Information to clearly discuss this point. We thank the referee for this nice suggestion that help improve our manuscript.

6) "First, for sensing applications, we believe the V-B defects in our isotopically purified host $h^{10}B^{15}N$ outperform conventional naturally abundant hBN_{nat} in all aspects." Would be more appropriate to specify aspects you demonstrated improvement. Are you also suggesting enhanced pressure sensing, if so why?

We thank the referee for the suggestion. We totally agree that claiming that $h^{10}B^{15}N$ outperforming hBN_{nat} in all aspects for sensing applications would be a

little too strong since we only measure the AC/DC magnetic field sensitivity in our experiment. Following the referee's suggestion, we have revised this claim to only mention the magnetic field sensing.

At the same time, although we didn't explicitly demonstrate in our experiment, we indeed believe that the temperature and pressure sensitivity for V_B^- can be also improved by using our isotopically purified $h^{10}B^{15}N$. The reason is that sensing temperature and pressure rely on measuring how the center frequency of the V_B^- ESR spectra response to external changes of temperature and pressure [1]. With much narrower and less crowded ESR transitions from V_B^- in $h^{10}B^{15}N$, we are able to determine the shift of the center frequency with a greatly improved sensitivity, in the same way as magnetic field sensing. One caveat is that, the temperature/pressure dependent shift of the ESR frequency can be also isotope-dependent. Indeed, in our temperature dependent T_1 measurement, we also notice that the shift of ESR center frequency for $h^{10}B^{15}N$ sample is $\sim 10\%$ smaller than the hBN_{nat} sample, presumably coming from the heavier atom mass thus weaker spin-phonon coupling strength. Nevertheless, this $\sim 10\%$ smaller temperature susceptibility in $h^{10}B^{15}N$ can be easily made up by the much narrower linewidth of the ESR transitions.

7) There are significant portions of the manuscript which would greatly benefit from a bit of rewriting/reorganizing for clarity.

a) Mixing of experimental details/results/theory/interpretation can be hard to follow in some places. Take for instance the discussion of the T_1 improvement. This is attributed to the heavier N15 mass twice before ever presenting the results. But the only mention of whether this fits the acquired data that both hBN types increase monotonically in T_1 with decreasing temperature, and not if the heavier N15 mass fits the observed increase in T_1 specifically. The isotopically engineered hBN also has an average lighter B mass, is this insignificant compared to the nearest neighbor N15's?

b) Another example, the introduction section reads a bit more like a conclusion than any conclusion portion.

c) In general, the relevant discussions are included, but at times scattered in many places, or experimental details discussed multiple paragraphs before discussion of the experiments

results.

We thank the referee for another very constructive suggestion! We must apologize for any confusion caused by the organization of our original manuscript. Following the Referee 3's as well as the previous two referee's suggestion, we have made structural changes and added clarifications throughout our manuscript.

For a), we completely agree with the referee that our arrangement of the discussion and experimental details may disorient the readers. The referee is also correct that the $h^{10}B^{15}N$ has a lighter boron atom mass when comparing to hBN_{nat} , which we believe, as the referee has suggested, may have a much smaller effect compared to the nearest ^{15}N nuclei. However, we have to admit that determining what phonon modes and how the V_B^- centers couple to require extensive theoretical and experimental investigation which are not involved in our work. To this end, following the referee's suggestion, in the revised manuscript, we have reorganized the T_1 paragraph to first focus on the experimental observations, and leave the underlying physical mechanism as an open question for future studies.

For b), originally, we were hoping to highlight the main results to the readers at the introduction part, so that the readers could be immediately captivated by what is new in our work. And since our manuscript has contained several quite different experimental components (e.g. isotope engineering and characterization, enhanced sensitivity, and nuclear spin polarization and control), we were hoping to present a clear roadmap to the readers at the introduction part. However, we agree with the referee that this could lead to some potential confusions to the reader. Following the referee's suggestion, we have condensed the introduction part in the revised manuscript, and leave the details to the newly added conclusion section.

For c), we thank the referee for the suggestion and apologize for the scattered discussions/experimental details. Since our manuscript includes several quite distinct experimental components, it has been quite a challenge for us to combine the different parts into a coherent story. Following the referee's suggestions, we have tried our very best to reorganize/rewrite the revised manuscript. In

particular, we have now arranged each subsection to start with a theoretical expectation and design, and then present the experimental results, followed by discussions and interpretations. We have also added a few additional subtitles to the main text to better guide the readers.

We would like to thank the referee again for the great suggestion and hope that, with all the changes we have made, the updated manuscript now conveys a better-organized story to the readers.

8) Ref #4 Listed twice in first citations

We thank the referee for drawing our attention to this mistake and have deleted the duplicated citation.

9) Typo SI, 1.2 section title “defects”

We appreciate the referee for catching this typo and have corrected the section title.

-
- [1] A. Gottscholl, M. Diez, V. Soltamov, C. Kasper, D. Krauße, A. Sperlich, M. Kianinia, C. Bradac, I. Aharonovich, and V. Dyakonov, *Nature Communications* **12**, 4480 (2021).
 - [2] M. Huang, J. Zhou, D. Chen, H. Lu, N. J. McLaughlin, S. Li, M. Alghamdi, D. Djugba, J. Shi, H. Wang, et al., *Nature Communications* **13**, 5369 (2022).
 - [3] P. Kumar, F. Fabre, A. Durand, T. Clua-Provost, J. Li, J. Edgar, N. Rougemaille, J. Coraux, X. Marie, P. Renucci, et al., *Physical Review Applied* **18**, L061002 (2022).
 - [4] A. Healey, S. Scholten, T. Yang, J. Scott, G. Abrahams, I. Robertson, X. Hou, Y. Guo, S. Rahman, Y. Lu, et al., *Nature Physics* **19**, 87 (2023).
 - [5] R. Schirhagl, K. Chang, M. Loretz, and C. L. Degen, *Annual review of physical chemistry* **65**, 83 (2014).
 - [6] S. Vaidya, X. Gao, S. Dikshit, I. Aharonovich, and T. Li, *Advances in Physics: X* **8**, 2206049 (2023).
 - [7] T. Yang, N. Mendelson, C. Li, A. Gottscholl, J. Scott, M. Kianinia, V. Dyakonov, M. Toth, and I. Aharonovich, *Nanoscale* **14**, 5239 (2022).

- [8] R. Rizzato, M. Schalk, S. Mohr, J. C. Hermann, J. P. Leibold, F. Bruckmaier, G. Salvitti, C. Qian, P. Ji, G. V. Astakhov, et al., *Nature Communications* **14**, 5089 (2023).
- [9] R. Gong, G. He, X. Gao, P. Ju, Z. Liu, B. Ye, E. A. Henriksen, T. Li, and C. Zu, *Nature Communications* **14**, 3299 (2023).
- [10] J. E. Fröch, L. P. Spencer, M. Kianinia, D. D. Totonjian, M. Nguyen, A. Gottscholl, V. Dyakonov, M. Toth, S. Kim, and I. Aharonovich, *Nano Letters* **21**, 6549 (2021).
- [11] C. Qian, V. Villafañe, M. Schalk, G. V. Astakhov, U. Kentsch, M. Helm, P. Soubelet, N. P. Wilson, R. Rizzato, S. Mohr, et al., *Nano Letters* **22**, 5137 (2022).
- [12] N. Mendelson, R. Ritika, M. Kianinia, J. Scott, S. Kim, J. E. Fröch, C. Gazzana, M. Westerhausen, L. Xiao, S. S. Mohajerani, et al., *Advanced Materials* **34**, 2106046 (2022).
- [13] B. Whitefield, M. Toth, I. Aharonovich, J.-P. Tetienne, and M. Kianinia, *Advanced Quantum Technologies* p. 2300118 (2023).
- [14] F. Zhou, Z. Jiang, H. Liang, S. Ru, A. A. Bettiol, and W. Gao, *Nano Letters* (2023).
- [15] X. Gao, S. Vaidya, K. Li, P. Ju, B. Jiang, Z. Xu, A. E. L. Allcca, K. Shen, T. Taniguchi, K. Watanabe, et al., *Nature Materials* **21**, 1024 (2022).
- [16] T. Clua-Provost, A. Durand, Z. Mu, T. Rastoin, J. Fraunié, E. Janzen, H. Schutte, J. Edgar, G. Seine, A. Claverie, et al., *arXiv preprint arXiv:2307.06774* (2023).
- [17] A. Haykal, R. Tanos, N. Minotto, A. Durand, F. Fabre, J. Li, J. Edgar, V. Ivady, A. Gali, T. Michel, et al., *Nature Communications* **13**, 4347 (2022).
- [18] E. Janzen, H. Schutte, J. Plo, A. Rousseau, T. Michel, W. Desrat, P. Valvin, V. Jacques, G. Cassabois, B. Gil, et al., *arXiv preprint arXiv:2306.13358* (2023).
- [19] M. Cambria, A. Norambuena, H. Dinani, G. Thiering, A. Gardill, I. Kemeny, Y. Li, V. Lordi, Á. Gali, J. Maze, et al., *Physical Review Letters* **130**, 256903 (2023).
- [20] S. Mondal and A. Lunghi, *npj Computational Materials* **9**, 120 (2023).

REVIEWERS' COMMENTS

Reviewer #1 (Remarks to the Author):

This is the second time I have reviewed this manuscript. In my first review, I wrote "...in my opinion the demonstrated gains in coherence and the achieved degree of nuclear spin polarization and control are fairly modest and are not nearly large enough to impact whether or not these defects can be used for any of the applications the authors list (quantum sensing, quantum registers)." I also asked a number of specific questions regarding important experimental details and parameters that were not included.

The authors have revised their work in response to my comments and questions and the reviews of the other referees, and I feel it is significantly improved as a result. I appreciate all the added measurements and information, as well as the thoughtful and friendly response in my reply to my comments. I now feel these results would be appropriate for publication in Nature Communications.

However, I still have two concerns that prevent me from recommending publication of this manuscript in its present form, and I recommend the authors revise the manuscript to address these points:

1) In the abstract the authors write: "For quantum registers, the individual addressability of the VB-hyperfine levels enables the dynamical polarization and coherent control of the three nearest-neighbor ^{15}N nuclear spins." They make similar comments regarding the relevance of their results for quantum registers in the main text. I still feel this is dramatically overselling these results. These are ensemble measurements, and the demonstrated coherence times and degree of nuclear spin polarization are so far below state of the art in other systems that in my opinion they are essentially irrelevant to any application that requires a "quantum register." It is one thing to mention "quantum registers" as a general motivation, and another to imply that these results somehow enable applications that require quantum registers. I strongly recommend the authors simply present their results and allow experts to gauge how useful these results will be for any specific applications that are not demonstrated here.

2) In their response to my comment that the isotopic mixture demonstrated here may not necessarily be optimal, the authors wrote in their response that a prior work (reference 27 in the manuscript): "...has shown that VB- in $h^{10}\text{B}^{14}\text{N}$ and $h^{11}\text{B}^{14}\text{N}$ did not exhibit a clear improvement in spin echo T_2 compared to natural abundant hBN." While this important context has now also been included in the revised manuscript in 2.2 of the Methods section, in the main text this paper is only

cited in the context of dynamical decoupling: "To this end, prior experimental works have focused on designing dynamical decoupling sequences to isolate spin defects from the local nuclear spin environment in hBN [24, 27, 28]." I think this is somewhat misleading, and that this work, as well as any prior works on defects in isotopically enriched hBN should be explicitly discussed in the introduction of the main text to give these current results their proper context.

Reviewer #2 (Remarks to the Author):

Spin defects in wide bandgap semiconductors are regarded as promising candidates for quantum technologies such as spin qubits and quantum sensors working at room temperature. Especially, since spin defects in 2D materials can be applied to flexible applications, the topic of this manuscript is suitable for nature communications, I think. The isotope engineering is an important way to improve spin properties. Actually, nitrogen-vacancy in diamond as well as silicon vacancy in SiC can achieve longer spin coherence time by isotope engineering. For hBN, this study firstly showed the improvement of spin properties for spin defect (Vb) in hBN by isotope engineering. In principle, isotope engineering works for any spin defects. However, in reality, it is important to observe such improvement by experiments. The results shown in the manuscript is well understood because authors obtained those results by careful/accurate measurements. The answers to my comments from the authors are reasonable and I understand now. So, I would suggest that the current version of the manuscript can be accepted for publication.

Takeshi Ohshima

Reviewer #3 (Remarks to the Author):

I believe the authors have done a good job of addressing my technical concerns, have improved the structure of the manuscript, and added very informative additional data and discussions. I also believe that the authors have sufficiently addressed the concerns of other reviewers, specifically reviewer 1, thereby improving the work. I believe the manuscript is suitable for publication in Nature Communications.

We would like to sincerely thank all the referees for the careful reviews and the support of publications.

Reviewers Comments:

Reviewer #1 (Remarks to the Author):

This is the second time I have reviewed this manuscript. In my first review, I wrote “...in my opinion the demonstrated gains in coherence and the achieved degree of nuclear spin polarization and control are fairly modest and are not nearly large enough to impact whether or not these defects can be used for any of the applications the authors list (quantum sensing, quantum registers).” I also asked a number of specific questions regarding important experimental details and parameters that were not included.

The authors have revised their work in response to my comments and questions and the reviews of the other referees, and I feel it is significantly improved as a result. I appreciate all the added measurements and information, as well as the thoughtful and friendly response in my reply to my comments. I now feel these results would be appropriate for publication in Nature Communications.

We must thank the referee for reviewing our revised manuscript for the second time and being supportive of publication. We are glad to hear that our revisions following referees’ constructive suggestions have improved our manuscript.

However, I still have two concerns that prevent me from recommending publication of this manuscript in its present form, and I recommend the authors revise the manuscript to address these points:

1) In the abstract the authors write: “For quantum registers, the individual addressability of the VB- hyperfine levels enables the dynamical polarization and coherent control of the three nearest-neighbor ^{15}N nuclear spins.” They make similar comments regarding the relevance of their results for quantum registers in the main text. I still feel this is dramatically overselling these results. These are ensemble measurements, and the demonstrated coherence times and degree of nuclear spin polarization are so far below state of the art in other systems that in my opinion they are essentially irrelevant to any application that

requires a "quantum register." It is one thing to mention "quantum registers" as a general motivation, and another to imply that these results somehow enable applications that require quantum registers. I strongly recommend the authors simply present their results and allow experts to gauge how useful these results will be for any specific applications that are not demonstrated here.

We thank the referee for bringing up a very good point and agree that the current wording of the quantum register applications may be too strong. Therefore, we have deleted/revised relevant wordings so that the improved nuclear spin control is presented without this implication.

2) In their response to my comment that the isotopic mixture demonstrated here may not necessarily be optimal, the authors wrote in their response that a prior work (reference 27 in the manuscript): "...has shown that VB- in $h^{10}B^{14}N$ and $h^{11}B^{14}N$ did not exhibit a clear improvement in spin echo T2 compared to natural abundant hBN." While this important context has now also been included in the revised manuscript in 2.2 of the Methods section, in the main text this paper is only cited in the context of dynamical decoupling: "To this end, prior experimental works have focused on designing dynamical decoupling sequences to isolate spin defects from the local nuclear spin environment in hBN [24, 27, 28]." I think this is somewhat misleading, and that this work, as well as any prior works on defects in isotopically enriched hBN should be explicitly discussed in the introduction of the main text to give these current results their proper context.

We apologize for not discussing this important context more explicitly in our manuscript. To this end, we have added a sentence in the introduction: "We note that a prior study has investigated the effect of changing only boron isotopes in hBN and do not observe significant improvements in the spin properties of V_B^- " and cited the previous work. We hope that this addition in the main text would make the context clearer for the audience.

Reviewer #2 (Remarks to the Author):

Spin defects in wide bandgap semiconductors are regarded as promising candidates for quantum technologies such as spin qubits and quantum sensors working at room temperature. Especially, since spin defects in 2D materials can be applied to flexible applications, the topic of this manuscript is suitable for nature communications, I think. The isotope engineering is an important way to improve spin properties. Actually, nitrogen-vacancy in diamond as well as silicon vacancy in SiC can achieve longer spin coherence time by isotope engineering. For hBN, this study firstly showed the improvement of spin properties for spin defect (Vb) in hBN by isotope engineering. In principle, isotope engineering works for any spin defects. However, in reality, it is important to observe such improvement by experiments. The results shown in the manuscript is well understood because authors obtained those results by careful/accurate measurements. The answers to my comments from the authors are reasonable and I understand now. So, I would suggest that the current version of the manuscript can be accepted for publication.

Takeshi Ohshima

We sincerely thank the referee for the careful reading of our work and the support for publication!

Reviewer #3 (Remarks to the Author):

I believe the authors have done a good job of addressing my technical concerns, have improved the structure of the manuscript, and added very informative additional data and discussions. I also believe that the authors have sufficiently addressed the concerns of other reviewers, specifically reviewer 1, thereby improving the work. I believe the manuscript is suitable for publication in Nature Communications.

We are sincerely grateful for the referee's comprehensive review of our manuscript and support of our work!